# DATA SELECTION FOR FINE-TUNING VISION LANGUAGE MODELS VIA CROSS MODAL ALIGNMENT TRAJECTORIES

## ABSTRACT

Data-efficient learning aims to eliminate redundancy in large training datasets by train-ing models on smaller subsets of the most informative examples. While data selection has been extensively explored for vision models and large language models (LLMs), it remains underexplored for Large Vision-Language Models (LVLMs). Notably, none of existing methods can outperform random selection at different subset sizes. In this work, we propose the first principled method for data-efficient instruction tuning of LVLMs. We prove that examples with similar cross-modal attention matrices during instruction tun-ing have similar gradients. Thus, they influence model parameters in a similar manner and convey the same information to the model during training. Building on this insight, we propose XMAS, which clusters examples based on the trajectories of the top singu-lar values of their attention matrices obtained from fine-tuning a small proxy LVLM. By sampling a balanced subset from these clusters, XMAS effectively removes redundancy in large-scale LVLM training data. Extensive experiments show that XMAS can discard 50% of the LLaVA-665k dataset and 85% of the Vision-Flan dataset while fully preserving per-formance of LLaVA-1.5-7B on 10 downstream benchmarks and speeding up its training by $1.2\times$. This is 30% more data reduction compared to the best baseline for LLaVA-665k.

## 1  INTRODUCTION

Large Vision-Language Models (LVLMs) have demonstrated impressive capabilities in understanding and reasoning over multimodal inputs (Liu et al., 2023; 2024a; OpenAI, 2023). LVLMs need to be trained on large data to obtain satisfactory performance. However, the amount of information in large datasets does not scale linearly with their size, due to redundancy (Sorscher et al., 2022). This raises the following key question: *can we eliminate redundancy in large training datasets of LVLMs without harming their performance?* Answering this question enables efficient training and guides data collection.

There has been a lot of recent efforts in developing data-efficient methods for training foundation models. For Large Language Models (LLMs), heuristic metrics such as middle perplexity (Marion et al., 2023), high learnability (Zhou et al., 2023b), large gradient norm (El2N) Paul et al. (2021), and highest uncertainty (Bhatt et al., 2024; Maharana et al., 2023) are commonly used. Other heuristics remove duplicates (Abbas et al., 2023) or select central examples in the embedding space (Bhatt et al., 2024). For LVLMs, heuristics based on CLIP-Score (Gadre et al., 2023; Chen et al., 2024), influence function (Liu et al., 2023), or sampling from activation clusters of carefully chosen layers (Lee et al., 2024) have been proposed. Notably, none of existing methods outperform random selection at various subset sizes, as we confirm in our experiments.

In this work, we address this problem from an optimization perspective, by studying the effect of every example on minimizing the training loss. As machine learning models are trained with gradient methods, examples that have similar gradients during the training affect the model parameters in a similar manner.

Hence, redundancy for training should be defined as gradient similarity (Mirzasoleiman et al., 2020). However, identifying examples with similar gradients during training becomes very challenging for LVLMs. First, LVLMs have billions of parameters and gradient similarity in such a high-dimensional space becomes vacuous and prohibitively expensive to calculate. Besides, as gradients change during the training, one should take into account the similarity between high-dimensional gradients during the entire training process. Finally, image and text embeddings lie in different spaces, creating a gap between the modalities: the distances among image embeddings, text embeddings, and image–text embeddings have different magnitudes. This phenomenon has also been observed in prior work (Yi et al., 2024; Role et al., 2025). As a result, the part of the gradient (captured by attention) corresponding to cross-modal alignment has a different magnitude than the part corresponding to individual modalities. This makes similarity calculation based on full gradients ineffective.

In this work, we address the above challenges and propose a theoretically-rigorous and efficient method to eliminate redundancy in large training datasets of LVLMs. First, we analyze a single-layer transformer and prove that the pairwise gradient distance between examples at a checkpoint can be upper-bounded by the distance between their cross-modal attention matrices. Then, we show that for instruction tuning where the Hessian is small, examples that have bounded distance between their cross-modal attention matrices at two checkpoints have bounded gradient distance between the checkpoints. Finally, we propose Cross Modal Alignment SVD (XMAS) that fine-tunes a small proxy VLM and tracks the trajectory of largest singular values of cross-modal attention matrices of examples at a few checkpoints during the training. XMAS finds examples with similar gradients by clustering attention trajectories. Top singular values of a matrix are closely related to its norm. So, examples that have similar top singular values for their cross-modal attention matrices have similar gradient norms (i.e. amount of cross-modal alignment). XMAS finds clusters of examples with similar gradient norms throughout the entire training. Within every cluster, examples are learned together (at a similar pace) and thus have similar learning dynamics. Every cluster contains subgroups of examples with similar gradient vectors (directions) with slightly different alignment trajectory patterns. By sampling examples with the most stable alignment trajectory, XMAS selects the central example from every gradient subgroup. Selecting a balanced subset of the most stable examples from alignment trajectory clusters eliminates redundancy and ensures superior performance on any unseen downstream task. We also theoretically analyze the convergence of training on XMAS subsets.

Our experiments demonstrate that XMAS outperforms existing baselines at various data budgets and can discard 50% of the LLaVA-665k dataset and 85% of the Vision-Flan dataset while fully preserving performance of LLaVA-1.5-7B on 10 downstream benchmarks, and speeding up its instruction tuning (including the time for data selection) by $1.2\times$. This is 30% more data reduction compared to the best baseline for LLaVA-665k. We also conduct an extensive ablation study on different components of our method.

## 2 RELATED WORKS

High-quality data is crucial for ensuring satisfactory performance of LLMs and LVLMs.

**Data-efficient Training of LLMs.** For instruction tuning, manually crafted high-quality instruction/response pairs was shown highly effective (Zhou et al., 2023a). Motivated by this, several studies explored using LLMs such as ChatGPT, or training on textbooks (Eldan & Li, 2023; Li et al., 2023c; Chen et al., 2023). Metrics such as diversity (Bukharin & Zhao, 2023; Du et al., 2023; Tirumala et al., 2023), difficulty (Bhatt et al., 2024; Marion et al., 2023; Zhou et al., 2023a), middle perplexity rankings (Marion et al., 2023), high gradient norm (EL2N) (Paul et al., 2021), memorization ranking (Biderman et al., 2023), and high learnability (difference between initial and final loss values) (Zhou et al., 2023b) have been explored. However, these methods assign similar scores to similar examples and thus cannot eliminate redundancy. To address this, SemDeDup (Abbas et al., 2023) removes redundancy by clustering embeddings. D2 Pruning (Maharana et al., 2023) prunes data using graph-based message passing to balance diversity and difficulty.

Despite being effective for LLMs, these methods perform poorly for LVLMs and are often outperformed by random selection, as we will confirm in our experiments.

**Data-efficient Training of LVLMs.** There has been recent efforts for selecting high-quality multimodal data. CLIP-Score (Gadre et al., 2023) selects examples with highest image–text similarity based on a pretrained CLIP model. However, CLIP-Score overlooks question–answer relevance and thus performs poorly for LVLMs. Self-Sup (Sorscher et al., 2022) clusters embeddings and selects examples closest to cluster centroids. SELF-FILTER trains a scoring model along with the VLM to learn difficulty of training instructions based on feature extracted from CLIP and GPT4V. Then, it uses the scoring model to select challenging instructions and filters them for diversity (Chen et al., 2024). TIVE (Liu et al., 2024c) selects examples that have the largest gradient similarity (influence) to other examples in the same task and selects more from tasks with smallest average influence. SELF-FILTER and TIVE are very expensive and yield suboptimal performance. Most recently, COINCIDE (Lee et al., 2024) proposed to train a proxy LVLM and cluster examples based on activations of carefully selected layers, and sampling more from clusters that are closer to each other and less from denser clusters (Lee et al., 2024). However, none of existing methods outperform random selection at various subset sizes, as we will confirm in our experiments.

**Targeted Data Selection.** Targeted data selection methods such as LESS (Xia et al., 2024) and ICONS (Wu et al., 2024) select influential training samples with largest gradient similarity to a validation set. Targeted data selection approaches have two major limitations: (i) they require computing gradients for every training example, which is even more computationally expensive than directly fine-tuning on the full dataset; and (ii) they rely on a validation set, which is often unavailable or impractical when training models intended for a broad range of downstream tasks. In our work, we do not assume access to a validation data.

## 3 PROBLEM FORMULATION

**Large Vision Language Model (LVLM).** An LVLM consists of a vision encoder, an LLM, and a projector. We denote all the model parameters by $\phi_{all}$. An input $(v^i, t^i, y^i)$ to LVLM consists of an image $v^i$, an instruction $t^i$ and the corresponding answer $y^i$. The encoded image is projected to the language space using the projector, before being concatenated with the instruction and fed into the language model. The response $y^i$ is then sampled from the following conditional probability distribution:

$$p_\phi(y^i|v^i, t^i) = \prod_{j \in V} p_{\phi_{all}}(y^i_j|v^i, t^i, y^i_{<j}). \tag{1}$$

where $y^i_j$ denotes the token at index $j$ and $y^i_{<j}$ denotes all tokens before index $j$.

**Visual Instruction Tuning (VIT).** To adapt a pre-trained LVLM for following specialized task instructions, visual instruction tuning (VIT), which is a type of supervised fine-tuning (SFT), is employed on a dataset $\mathcal{D}_{VIT} = \{(v, t, y)^i\}_{i \in V}$. During instruction-tuning, the vision encoder is kept frozen and only the projector and the LLM are trained. We use $\phi$ to denote the trainable parameters. Therefore, the visual instruction tuning objective is to minimize the following negative log likelihood loss:

$$\min_\phi \mathcal{L}(\phi, \mathcal{D}_{VIT}) = -\frac{1}{|V|} \sum_{(v,t,y)^i \in \mathcal{D}_{VIT}} \left[ \log p_\phi(y^i|v^i, t^i) \right] \tag{2}$$

In practice, gradient methods are applied to train the model by minimizing the above loss function.

**Finding Redundant Clusters in the VIT Data.** Consider an LVLM with parameters $\phi$. Our goal is to find a clustering of the data $V = \{C_1 \cup \cdots \cup C_K\}$ such that in every cluster $C_k$ examples have similar gradients during the entire instruction tuning process. Formally, let $\Phi$ be the set of trainable parameters of the model during fine-tuning. Then we wish to find a solution to the following problem:

$$V = \{C_1 \cup \cdots \cup C_K\} \quad \text{s.t.} \quad \max_{\phi \in \Phi} \|\nabla \mathcal{L}_i(\phi) - \nabla \mathcal{L}_j(\phi)\| \leq R \quad \forall i, j \in C_k, \forall k \in [K], \tag{3}$$

where $\nabla \mathcal{L}_i(\phi)$ is the gradient of example $i$ at parameter $\phi$. Examples in every cluster have similar gradients during instruction tuning and hence are redundant w.r.t each other for training.

**Sampling a Balanced Subset From Clusters.** Having the above clustering, for a given data budget $B$, we sample a balanced (i.e., same number of examples from each cluster) subset of examples $S \subseteq V$ from all the clusters $\{C_1 \cup \cdots \cup C_K\}$ to train the target model. In doing so, we effectively eliminate redundancy in the multimodal data, and ensure satisfactory performance on various (unseen) downstream tasks.

## 4 METHOD: EXTRACTING NON-REDUNDANT SUBSETS OF THE VIT DATA

Solving Eq. 3 is very challenging as it requires training the model, saving the gradients of all the training examples after every parameter update, and clustering the concatenated gradient vectors. However, for LVLMs with billions of parameters, this becomes infeasible and does not improve the training efficiency.

To address this, we wish to train a small proxy VLM and use its training dynamics to approximate the pairwise gradient distances during instruction tuning. If this can be done, one can cluster the training examples based on the estimated gradient distances obtained from the proxy, and randomly sample a balanced subset from the clusters to train the larger target LVLM on the non-redundant subset. If the proxy model is small-enough, training the proxy to find the non-redundant subset and instruction tuning the larger target LVLM will be faster than instruction tuning on the full data.

### 4.1 FINDING CLUSTERS OF EXAMPLES WITH SIMILAR GRADIENT

In this section, we wish to answer the following question: when fine-tuning a proxy VLM on the instruction-tuning data, which statistics can be used to upper-bound pairwise gradient distances during the training?

Answering the above question requires understanding the mechanism by which VLMs learn from the instruction-tuning data. Intuitively, instruction tuning aligns the vision and language modalities and enables the LLM to understand the content of the images. This is primarily done via the trainable attention matrices in the LLM structure. Formally, consider the $l$-th layer of the language decoder of a VLM with parameter $\phi$ with hidden dimension size $D$. The per-layer attention matrix for example $i \in V$ is defined as:

$$A_i^l(\phi) = \text{softmax}\left(\frac{Q_l \otimes K_l^T}{\sqrt{D}}\right) \in \mathbb{R}^{N \times N}, \tag{4}$$

where $N = n_I + n_T$ which is the total number of image and text tokens, and $Q_l, K_l \in \mathbb{R}^{N \times D}$ are the concatenated query and key matrices along the hidden dimension across all the attention heads. $A_i^l(\phi)$ consists of both the cross-modal attention and intra-modal attention terms. The cross-modal attention part $\chi_i^l(\phi) \in \mathbb{R}^{n_T \times n_I}$ corresponds to the bottom left block of of the per-layer attention matrix $A_i^l(\phi)$.

**Definition 4.1** (Cross-modal alignment score $\sigma$). *For data point $i$, we define $\sigma_i(\phi)$ as the sum of the top five singular values of its cross-modal attention matrix $\chi_i(\phi) = \sum_{l=1}^{L} \chi_i^l(\phi)$. Intuitively, $\sigma$ captures the amount of cross-modal alignment for individual examples.*

**Remark.** Cross-modal attention matrices are transferrable between proxy and target LVLMs (Zhao et al., 2024). This implies that cross-modal alignment scores obtained based on a proxy VLM closely estimates alignment for the larger LVLM. We will empirically confirm this observation in our ablation studies.

#### 4.1.1 BOUNDING GRADIENT DISTANCE VIA ATTENTION DISTANCE

Next, we analyze a single-layer transformer with one attention head and RMS layer normalization, trained using the Frobenius norm squared loss. This setting has been studied in several recent theoretical analysis

of transformers (Ormaniec et al., 2024; Song et al., 2024). RMS layer normalization is standard practice in many recent open-source models, such as LLaMA and Mistral. In practice, the gain parameter $g$ of RMS normalization is often initialized to a small value to help stabilize early training, thereby preventing activation blowups and facilitating stable learning in residual architectures Zhang & Sennrich (2019).

The following theorem shows that at every step $t$ during the training, pairwise attention distances can be used to upper-bound pairwise gradient distances.

**Theorem 4.1.** *Consider a single-layer transformer with a single attention head with RMS layer normalization, that is trained using the Frobenius norm squared loss. Let $D$ be the hidden dimensionality of the model, $N$ be the number of input tokens, and $c \geq \|\phi^t\|$ be the upper-bound on the norm of model parameters. Then, if the gain $g$ of RMS normalization layer is small enough $g < N^{-5/8}D^{-1/8}c^{-3/4}$ and for all examples $p \in V$ the distance between cross-modal attention matrices of the proxy and target models are bounded, i.e., $\|\chi_p(\phi_{proxy}^t) - \chi_p(\phi_{target}^t)\|_F \leq T$, then pairwise attention distances of the proxy model $K_{ij}^t = \|\chi_i(\phi_{proxy}^t) - \chi_j(\phi_{proxy}^t)\|_F$ for examples $i, j \in V$ dominate the bound on their pairwise gradient distance of the target model at every step $t$ in training:*

$$\|\nabla\mathcal{L}_i(\phi_{target}^t) - \nabla\mathcal{L}_j(\phi_{target}^t)\|_F \leq \frac{4}{\sqrt{3}} \cdot (K_{ij}^t + 2T) + \frac{8\sqrt{N}}{3\sqrt{3}c} = \Delta_{ij}^t \tag{5}$$

All the proofs can be found in Appendix A.

**Remark.** Under the assumptions of Theorem 4.1, we have $K_{ij}^t \leq 2\sqrt{N}$. Thus, for a good proxy model with small $T$, i.e. $T \ll \sqrt{N}$, $K_{ij}^t$ dominates the upper-bound on pairwise gradient distances of the target model. Hence, clustering based on $K_{ij}^t$ is similar (up to some error) to clustering based on gradients of the target model at step $t$.

### 4.1.2 BOUNDING GRADIENT DISTANCE THROUGHOUT FINETUNING

Theorem 4.1 shows that examples with similar cross-modal alignment score at a particular step $t$ during training have similar gradients at step $t$. However, our goal is to find groups of examples with similar gradient *throughout* the training. Next, we show that for fine-tuning where loss has a small bounded curvature (Gekhman et al., 2024; Yang et al., 2024), examples that have similar cross-modal attention matrices two checkpoints also have similar gradients *between* those checkpoints.

**Theorem 4.2.** *Under the assumptions of Theorem 4.1, Suppose the per-example loss during fine-tuning admits a second-order Taylor approximation with bounded curvature, i.e., $\|\nabla^2\mathcal{L}_i(\phi_{target}^t)\| \leq \beta \ \forall t$, Then, for any two checkpoints $\phi^{t_1}, \phi^{t_2}$ where $\|\phi^{t_1} - \phi^{t_2}\| \leq \delta$, the largest pairwise attention distance between them provides an upper bound on the gradient distance at any intermediate checkpoint. Specifically, for all $t_z \in [t_1, t_2]$, we have:*

$$\|\nabla\mathcal{L}_i(\phi_{target}^{t_z}) - \nabla\mathcal{L}_j(\phi_{target}^{t_z})\|_F \leq \max\{\Delta_{ij}^{t_1}, \Delta_{ij}^{t_2}\} + 2\delta\beta = \Delta_2 \tag{6}$$

**Remark.** The above theorem implies that if two examples have similar attention matrices at a few checkpoints during instruction tuning of the proxy model, then examples have similar target gradients *throughout* instruction tuning. Since curvature $\beta$ is small during fine-tuning, $\max\{\Delta_{ij}^{t_1}, \Delta_{ij}^{t_2}\} \leq \frac{8\sqrt{N}}{\sqrt{3}}$ will dominates the upper-bound of gradient distance in Eq 6. Based on this, we define alignment trajectory:

**Definition 4.2** (Alignment trajectory.)**.** *Consider $r$ checkpoints $\{\phi^{t_1}, \phi^{t_2}, \ldots, \phi^{t_r}\}$ during fine-tuning a proxy model on $\mathcal{D}_{VIT}$. The cross-modal alignment trajectory of example $i \in V$ is defined as:*

$$T_i = \{\sigma_i(\phi^{t_1}), \sigma_i(\phi^{t_2}), \ldots, \sigma_i(\phi^{t_r})\} \tag{7}$$

*where $\sigma_i(\phi^{t_j})$ represents the cross-modal alignment score of example $i$ at checkpoint $\phi^{t_j}$.*

---

**Algorithm 1** Data Selection Based on Cross-Modal Alignment Trajectories (XMAS)

---

**Require:** Training dataset $\mathcal{D}_{\text{VIT}}$, a fixed data budget $B$, number of clusters $K$
**Ensure:** Subset $S \subseteq \mathcal{D}_{\text{VIT}}$, $|S| \leq B$
1: $S \leftarrow \emptyset$
2: Train a small proxy VLM and cluster examples in $\mathcal{D}_{\text{VIT}}$ based on their alignment trajectories
3: Compute Instability score $S_i$ for each example in $\mathcal{D}_{\text{VIT}}$ according to Eq 8
4: Sort clusters by size to get $C = \{C_1, C_2, \ldots, C_K\}$
5: **for** $k = 1$ to $K$ **do**
6:     **if** $|C_k| \leq R_k = \frac{B - |S|}{K - k + 1}$ **then**
7:         $S \leftarrow S \cup C_k$
8:     **else**
9:         $S \leftarrow S \cup C'_k$ where $C'_k \subset C_k$ is the subset of $R_k$ most stable examples
10: **return** $S$

---

**Clustering alignment trajectories.** Having cross-modal alignment trajectories for all examples in the dataset using the proxy model, we cluster these trajectories using the $K$-means clustering. In doing so, we get clusters of examples with similar gradients during the training $\{C_1, C_2, \ldots, C_K\}$.

### 4.2 SAMPLING A BALANCED SUBSET FROM GRADIENT CLUSTERS

Next, we sample a balanced subset from the alignment trajectory clusters to eliminate redundancy. While random sampling from the clusters is already effective, sampling examples with more stable (less oscillating) trajectories yields a better performance in practice.

**Definition 4.3** (Instability score). *The instability score of example $i \in V$ is the total oscillation in the cross-modal alignment score of $i$ during fine-tuning:*

$$S_i = \sum_{j=1}^{T} |\sigma_i(\phi^{t_j}) - \sigma_i(\phi^{t_{j-1}})|. \tag{8}$$

Intuitively, examples with smallest instability score within every cluster are centers of subgroups in that cluster. Sampling examples with smallest instability score ensures selecting a diverse set of representative examples from the clusters. We will confirm the effectiveness of stability sampling in our ablation studies.

### 4.3 DATA SELECTION BASED ON CROSS MODAL ALIGNMENT TRAJECTORIES (XMAS)

To summarize, our method, XMAS includes three main steps. First, we fine-tune a proxy model to get the cross-modal alignment trajectories (see Def. 4.2) for all the examples. Then, we cluster these trajectories to find examples with bounded gradient difference throughout instruction tuning. Finally, we sample a balanced subset of stable points from these clusters. The pseudocode of XMAS is given in Algorithm 1 and a visualization can be found in Figure 5 in the Appendix.

**Convergence of XMAS.** Finally, we analyze the convergence of finetuning target model on the subset. The following corollary shows that training on the subset selected by XMAS converges to a neighborhood of the optimal solution found by training the target LVLM on the full dataset. For brevity, we show the informal version of the corollary here and defer the formal statement of the corollary and its proof to Appendix A.3.

**Corollary 4.3** (Informal: Convergence of XMAS). *Under the assumptions of Theorem 4.2, applying incremental gradient methods with stepsize $\eta$ on subsets found by XMAS, converges to a neighborhood of the solution $\phi^*$ found by training on full data.*

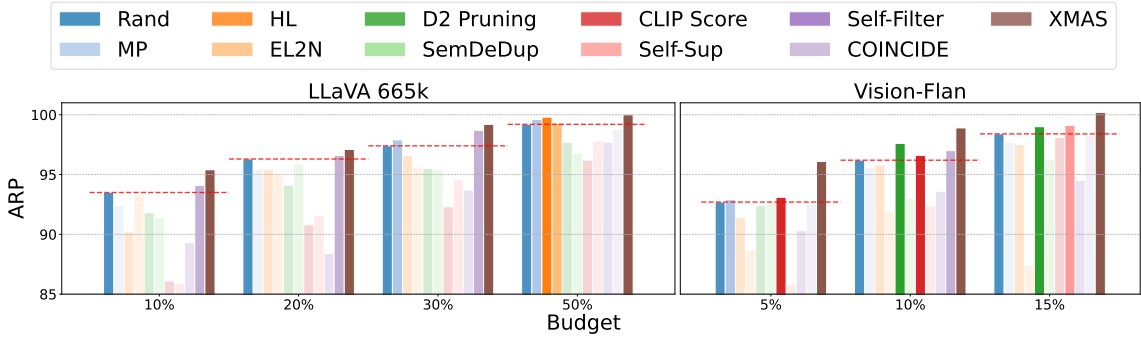

Figure 1: The average relative performance (ARP) of different subsets of (left) LLaVA 665k and (right) Vision-Flan when fine-tuning LLaVA-1.5-7B. Methods that outperform random selection are shown in opaque. XMAS is the only method that surpasses random selection across different budgets on both datasets. MP and HL finetune the target model on full data to find subsets and thus do not yield any speedup.

## 5 EXPERIMENTS

In this section, we first describe our experimental settings, followed by an empirical evaluation of our method on two widely used VIT datasets in Section 5.1. We then report run time of XMAS and analyze the impact of different design choices (proxy models, number of checkpoints, number of clusters, sampling strategies, and attention matrices) in our approach. Additional results and qualitative analysis are deferred to Appendix C.

**Models.** For the target LVLMs, we use the pre-trained LLaVA-1.5-7B model (Liu et al., 2024a). For the proxy model, we use the TinyLLaVA-2B (Zhou et al., 2024).

**VIT datasets.** We apply coreset selection to two separate VIT datasets: LLaVA 665k (Liu et al., 2024a) and Vision-Flan (Xu et al., 2024). LLaVA 665k comprises 665k VIT examples collected from 12 vision-language datasets. On the other hand, Vision-Flan consists of 191 vision-language tasks, each featuring approximately 1k expert-labeled VIT samples, amounting to a total of 186k instances.

**Training details.** In all experiments, we fine-tune the target models using LoRA (Hu et al., 2022) for one epoch, following the official finetuning hyperparameters specified in LLaVA-1.5. For proxy models, we train without LoRA for one epoch, following the official hyperparameters specified in TinyLLaVA. This results in a total of $T = 7$ checkpoints. For K-means, we set the number of clusters $K$ to 1000.

**Baselines.** We compare XMAS against 11 data selection baselines, including Random selection (**Rand**), Middle Perplexity (**MP**) (Marion et al., 2023), and Highest Learnability (**HL**) (Zhou et al., 2023b). In addition, we consider EL2N (Paul et al., 2021), Self-Sup (Sorscher et al., 2022), CLIP-Score (Hessel et al., 2021), SemDeDup (Abbas et al., 2023), D2 Pruning (Maharana et al., 2023). We also include two recent methods proposed for LVLMs: Self-Filter (Chen et al., 2024) and COINCIDE (Lee et al., 2024).

**Evaluation.** We evaluate the performance of the fine-tuned target models on reasoning, hallucinations, perception, and cognition capabilities. For all the experiments, we evaluate the aforementioned capabilities of the fine-tuned model on POPE (Li et al., 2023a), TextVQA (Singh et al., 2019), MME-Perception (Liang et al., 2024), ScienceQA (Lu et al., 2022), VizWiz (Gurari et al., 2018), MMBench (Liu et al., 2024b), LLaVABench (Liu et al., 2023), MMVet (Yu et al., 2023), VQAv2 (Goyal et al., 2017) and GQA (Hudson & Manning, 2019) datasets. We follow the same evaluation protocols outlined in LLaVA-1.5. We measure the relative performance of subsets as (subset performance / full data performance) × 100%. To compare between different methods, we **A**verage the **R**elative **P**erformance (ARP) across all evaluation datasets.

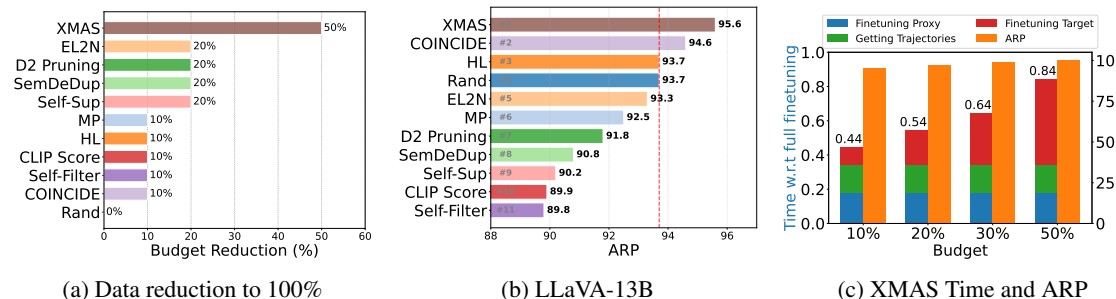

| (a) Data reduction to 100% | (b) LLaVA-13B | (c) XMAS Time and ARP |

Figure 2: (left) Data reduction to reach 100% Average relative performance (ARP) of LLaVA-1.5-7B on LLaVA 665k. XMAS obtains 30% more data reduction over the best baselines. (middle) ARP ranking of different 10% subsets of LLaVA 665k when fine-tuning LLaVA-1.5-13B. (right) ARP and ratio of total time (selection + training) w.r.t training target model on full dataset for XMAS at different budgets on LLaVA 665k. XMAS reduces training time by a factor of 0.84 (1.2× speedup) to reach 100% ARP.

## 5.1 MAIN RESULTS

**LLaVA 665k dataset.** As shown in Fig. 1 left, XMAS outperforms all baselines across different budgets. Relying on a single metric is worse than random selection for small budgets of 10-30% and worse than COINCIDE. This set of experiments confirms the observation made by (Lee et al., 2024) that single metrics lead to biased and redundant selection that, in turn, leads to sub-optimal results. Notably, XMAS reaches the performance of fine-tuning on the full datasets at only 50%, which is 2x data reduction compared to fine-tuning on the full dataset. This is 30% more data reduction over the best baselines as shown in Fig. 2a. When training LLaVA-13B on the subsets, Fig. 2b demonstrates that XMAS still achieves the highest ARP, demonstrating its effectiveness for training larger LVLMs.

**Vision-Flan dataset.** Fig. 1 right demonstrates the superior performance of XMAS, consistently outperforming random sampling by at least 2% while other baselines fail to surpass random at all budgets. Our method matches the full performance with only 15% of the total samples, which is 6.7x data reduction compared to fine-tuning on the full dataset.

## 5.2 ABLATION STUDIES AND ANALYSIS

**Computation time.** Fig. 2c reports the total time for data selection (proxy fine-tuning + trajectory extraction) and target model fine-tuning on LLaVA-665k. The clustering and sampling steps in our method incur negligible cost. We observe that XMAS at 50% data retains full-dataset performance while delivering a 1.2× speedup. Although COINCIDE's feature extraction is faster, it can only discard 10% of the data, resulting in higher fine-tuning costs for the target model and making it 1.1× slower than full training.

**Different proxy models.** To investigate the effect of the proxy model, we try two different scales of TinyLLaVA which are 0.5B and 2B. Fig. 3a compares the performance of COINCIDE and XMAS when varying the proxy models. XMAS outperforms COINCIDE for different proxy models.

**Number of checkpoints.** Fig. 3b illustrates the performance of XMAS for varying number of proxy checkpoints. Using 7 checkpoints covering one pass of the full dataset yields the best performance of 95.4%. While using only 1(3) checkpoint reduces the computation cost, it harms the performance by roughly 2(1)%. Notably, increasing the number of checkpoints to 11 not only increases the computation cost but also decreases the relative performance to 94.1%.

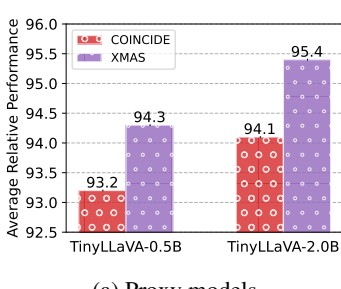
(a) Proxy models

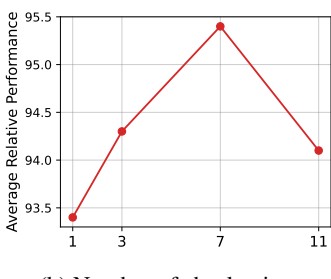
(b) Number of checkpoints

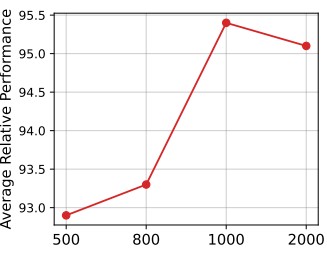
(c) Number of clusters

Figure 3: Average performance relative to full data for 10% subsets of LLaVA 665k found by COINCIDE and XMAS when varying (left) proxy model, (middle) number of checkpoints and (right) number of clusters.

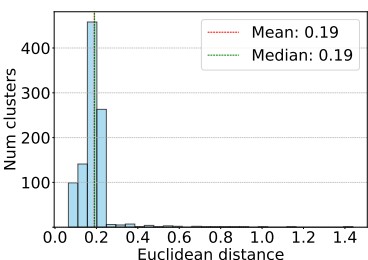

Figure 4: Per-cluster Euclidean distance between the cross-modal alignment trajectories of proxy model and target model on LLaVA 665k. Distance between trajectories of proxy and target models is very small.

Table 1: Average relative performances (ARP) over full data when training LLaVa-1.5-7B on 10% subsets of LLaVA 665K found by XMAS when using different (left) cluster sampling strategies and (b) attention matrices.

| Strategy | ARP |
|---|---|
| Random | 94.1 |
| Instability | **95.4** |

(a) Cluster sampling

| Attn matrix | ARP |
|---|---|
| Full | 94.0 |
| Cross-modal | **95.4** |

(b) Attention matrix

**Number of clusters.** Fig.3c illustrates the performance of XMAS for different numbers of clusters ($K$). Using 1,000 clusters yields the best performance of 95.4%. Increasing the number of clusters to 2,000 slightly lowers performance to 95.1% while fewer clusters (500 or 800) reduce performance to 92.9% and 93.3%.

**Proxy vs target trajectories.** Fig. 4 illustrates the per-cluster Euclidean distance between the alignment trajectories of proxy model (TinyLLaVA-2B) and target model (LLaVA-1.5-7B) on LLaVA 665k, i.e., $\sum_{i \in C_k} \|T_i^{\text{proxy}} - T_i^{\text{target}}\|_2 / |C_k|$. We see clearly that the trajectories of proxy and target models are similar as indicated by small Euclidean distance ($< 0.25$) for most of the clusters.

**Cluster sampling strategy.** Table 1a illustrates that cluster sampling based on the instability score performs better than random sampling by 1.3%. This validates our intuition in 4.2, which shows that sampling based on instability scores guides XMAS to select more representative examples.

**Choice of attention matrices.** Next, we study the effect of using the full vs cross-modal attention matrices. Table 1b shows that using only the cross-modal terms works better than using both intra-modal and cross-modal terms. This confirms that the cross-modal attention provides more informative signals than intra-modal for multi-modal data selection.

Additional ablation studies on the choice of alignment score, layer aggregation strategy, instability score, attention layers, and number of singular values are in Appendix C. Moreover, we show that COINCIDE is sensitive to the choice of layers used for feature extraction. Qualitative results of XMAS clusters are given in Fig. 7-10.

## 6 CONCLUSION

In this work, we present XMAS, an effective data selection method to improve data efficiency of visual instruction tuning for Large Vision Language Models (VLMs). By clustering examples based on their cross-modal alignment trajectories and sampling a balanced set of examples with the most stable trajectories from all the clusters, XMAS results in a significant reduction of the required training data without compromising the performance as compared to training on the complete dataset. Empirically, XMAS can drop 50% of LLaVA 665K and 85% of VisionFLAN datasets and train LLaVA-1.5-7B to the same accuracy of full data.

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

## A PROOFS

**Notations.** Consider a data set with $|\mathcal{D}_{\text{VIT}}|$ samples where each sample consists of $N = n_I + n_T$ number of tokens with embedding dimension D. We denote the dataset as $(X_i, y_i)_{i=1}^{\mathcal{D}_{\text{VIT}}}$, where $X_i \in \mathbb{R}^{N \times d}$, and $y_i \in \mathbb{R}^N$ is the label of the dataset. Following the settings in Ormaniec et al. (2024), we consider a single-layer attention Transformer model with a single head. Let $W_Q, W_K, W_V \in \mathbb{R}^{d \times D}$ denote the weight matrices of Q, K, and V projection matrices, respectively. The output from the Transformer model is formulated as:

$$A_i = \frac{X_i W_Q W_K^T X_i^T}{\sqrt{D}} \tag{9}$$

$$S_i = \text{Softmax}(A_i) = \text{Softmax}\left(\frac{X_i W_Q W_K^T X_i^T}{\sqrt{D}}\right) \tag{10}$$

$$F_i = S_i X_i W_V \tag{11}$$

where $D$ is the hidden dimension of the model.

We train the above Transformer using the Frobenius norm squared loss function.

$$\mathcal{L}_i = \frac{\|F_i - Y_i\|_F^2}{ND} \tag{12}$$

Note that, for brevity, we use $S_i$ instead of $A_i^l(\phi)$ in Equation 4 to denote the attention matrix. In addition, we write the cross-modal attention matrix $\chi_i(\phi)$ as $\chi_i$.

**Assumptions.** We assume that the model embedding $X_i$ and weight matrices $W_{\{Q,K,V\}}$ have bounded norms. Let $X, Q, K$, and $V$ be their corresponding upper bounds. In addition, the attention matrix $S_i$ also has a bounded norm $S$ as each entity is bounded between 0 and 1. Furthermore, we assume that the model always makes a reasonable prediction which is not far from the ground-truth label. In other words, the loss is bounded and we denote the upper bound of the norm of the difference between the predicted output and ground-truth label as $\alpha = \sup_i \|F_i - y_i\|$.

**Bound of target alignment trajectories.** From the assumption that the alignment trajectories of proxy and target models are close, we have:

$$\|\chi_i^{\text{proxy}} - \chi_i^{\text{target}}\| \leq T, \quad \forall i. \tag{13}$$

For $i$ and $j$ in the same cluster $C_k$ based on the proxy model, we have:

$$\|\chi_i^{\text{proxy}} - \chi_j^{\text{proxy}}\| \leq K_{ij}. \tag{14}$$

Using the triangle inequality:

$$\begin{aligned}
\|\chi_i^{\text{target}} - \chi_j^{\text{target}}\| &\leq \|\chi_i^{\text{target}} - \chi_i^{\text{proxy}}\| + \|\chi_i^{\text{proxy}} - \chi_j^{\text{proxy}}\| + \|\chi_j^{\text{proxy}} - \chi_j^{\text{target}}\| \\
&\leq 2T + K_{ij} = \epsilon'.
\end{aligned} \tag{15}$$

Therefore, for two samples $i$ and $j$ in the same cluster $C_k$ at any iteration $t$, we have:

$$\|\chi_i^{\text{target}} - \chi_j^{\text{target}}\| \leq \epsilon', \quad \forall t. \tag{16}$$

**Gradient decomposition.** Before bounding the gradient difference of the target model, we introduce the expressions of the model gradient w.r.t different weight matrices in the following lemma.

**Lemma 1.** *Jacobians of the attention weight matrices $W_{\{Q,K,V\}}$ have the following form:*

1. $\nabla_{W_V}\mathcal{L}_i(\phi^t) = \frac{2}{ND}(F_i - y_i)(S_iX_i \otimes I_D) = \frac{2}{ND}(S_iX_iW_V - y_i)(S_iX_i \otimes I_D)$

2. $\nabla_{W_Q}\mathcal{L}_i(\phi^t) = \frac{2}{ND}(S_iX_iW_V - y_i)(I_N \otimes W_V^T X_i^T)\frac{\partial S_i}{\partial A_i}\frac{X_i \otimes X_i W_K}{\sqrt{D}}$

3. $\nabla_{W_K}\mathcal{N}_i(\phi^t) = \frac{2}{ND}(S_iX_iW_V - y_i)(I_N \otimes W_V^T X_i^T)\frac{\partial S_i}{\partial A_i}(\frac{X_i \otimes X_i W_Q}{\sqrt{D}})\Lambda_{D,d}$

where $I_L$ denotes the identity matrix of size $L$ and $\Lambda_{D,d}$ is the commutation matrix.

**Lemma 2.** *Bound on the Frobenius norm of the Jacobian of the attention matrix:*

$$\|\nabla_{A_i}S_i\|_F \leq \frac{\sqrt{N}}{2} \tag{17}$$

## A.1 PROOF OF THEOREM 4.1

*Proof.* Since we have considered a simplified model with only three parameters, $W_Q$, $W_K$ and $W_V$, our goal now is to find $\|\nabla\mathcal{L}_i(\phi^t_{target}) - \nabla\mathcal{L}_j(\phi^t_{target})\|_F$ which can be expanded as

$$\|\nabla\mathcal{L}_i(\phi^t_{\text{target}}) - \nabla\mathcal{L}_j(\phi^t_{\text{target}})\|_F = \sqrt{\begin{array}{l}\|\nabla_{W_V}\mathcal{L}_i(\phi^t_{\text{target}}) - \nabla_{W_V}\mathcal{L}_j(\phi^t_{\text{target}})\|_F^2 \\ + \|\nabla_{W_Q}\mathcal{L}_i(\phi^t_{\text{target}}) - \nabla_{W_Q}\mathcal{L}_j(\phi^t_{\text{target}})\|_F^2 \\ + \|\nabla_{W_K}\mathcal{L}_i(\phi^t_{\text{target}}) - \nabla_{W_K}\mathcal{L}_j(\phi^t_{\text{target}})\|_F^2\end{array}} \tag{18}$$

To bound the LHS in Equation 18, we first bound each term in the RHS.

**Bound for $W_V$.** We simplify the first term in Equation 18 as follows:

$\|\nabla_{W_V}\mathcal{L}_i(\phi^t_{\text{target}}) - \nabla_{W_V}\mathcal{L}_j(\phi^t_{\text{target}})\|_F$

$\overset{(i)}{=} \frac{2}{ND}\|(S_iX_iW_V - y_i)(S_iX_i \otimes I_D) - (S_jX_jW_V - y_j)(S_jX_j \otimes I_D)\|_F$

$= \frac{2}{ND}\|S_iX_iW_V((S_iX_i - S_jX_j) \otimes I_D) + (S_iX_i - S_jX_j)W_V(S_jX_j \otimes I_D)$
$\quad + y_j((S_jX_j - S_iX_i) \otimes I_D) + (y_j - y_i)(S_iX_i \otimes I_D)\|_F$

$\overset{(ii)}{\leq} \frac{2}{ND}\|S_iX_iW_V((S_iX_i - S_jX_j) \otimes I_D)\|_F + \frac{2}{ND}\|(S_iX_i - S_jX_j)W_V(S_jX_j \otimes I_D)\|_F$
$\quad + \frac{2}{ND}\|y_j((S_jX_j - S_iX_i) \otimes I_D)\|_F + \frac{2}{ND}\|(y_j - y_i)(S_iX_i \otimes I_D)\|_F$

$\overset{(iii)}{\leq} \frac{2SXV}{N}\|(S_i(X_i - X_j) + (S_i - S_j)X_j\|_F + \frac{2SXV}{N}\|(S_i(X_i - X_j) + (S_i - S_j)X_j\|_F$
$\quad + \frac{2}{N}B\|(S_i(X_i - X_j) + (S_i - S_j)X_j\|_F + \frac{4BSX}{N}$

$\overset{(iv)}{\leq} \frac{8S^2X^2V}{N} + \frac{4SX^2V}{N}\|S_i - S_j\|_F + \frac{4BSX}{N} + \frac{2BX}{N}\|S_i - S_j\|_F + \frac{4BSX}{N}$

$\overset{(v)}{\leq} \left(\frac{4SX^2V}{N} + \frac{2BX}{N}\right)\|S_i - S_j\|_F + \frac{8SX}{N}(B + SXV) \tag{19}$

where (i) comes from Lemma 1; (ii) uses triangle inequality; (iii) applies assumed bounds; (iv) uses $\|X_i - X_j\|_F \leq 2X$; (v) rearranging.

Now, consider the following relation for the attention matrix:

$$S_i = \chi_i + \chi_i^T + S_i^r \tag{20}$$

where $\chi_i$ is the cross-modal attention part (bottom-left section) of the attention matrix that is considered in the actual experiments. $\chi_i$ is the same shape as $\chi_i$ with rest of the entries being zero. Similary, $S_i^r$ is the same shape as $S_i$, with entries in the cross-modal attention part being zero and all other entries being non-zero. Now, consider the following difference:

$$\|\nabla_{W_V}\mathcal{N}_i(\phi_{\text{target}}^t) - \nabla_{W_V}\mathcal{N}_j(\phi_{\text{target}}^t)\|_F$$

$$\stackrel{(i)}{\leq} \left(\frac{4SX^2V}{N} + \frac{2BX}{N}\right)\|\chi_i - \chi_j + \chi_i^T - \chi_j^T + S_i^r - S_j^r\|_F + \frac{8SX}{N}(B + SXV)$$

$$\stackrel{(ii)}{\leq} \left(\frac{8SX^2V}{N} + \frac{4BX}{N}\right)\|\chi_i - \chi_j\|_F + \left(\frac{4SX^2V}{N} + \frac{2BX}{N}\right)\|S_i^r - S_j^r\|_F + \frac{8SX}{N}(B + SXV)$$

$$\stackrel{(iii)}{\leq} \left(\frac{8SX^2V}{N} + \frac{4BX}{N}\right)\|\chi_i - \chi_j\|_F + \frac{4SX}{L}(4SXV + 3B)$$

$$= \left(\frac{8SX^2V}{N} + \frac{4BX}{N}\right)\epsilon' + \frac{4SX}{N}(4SXV + 3B) \tag{21}$$

where (i) from equation 20; (ii) applies triangle inequality; (iii) $\|S_i^r - S_j^r\|_F \leq 2S$.

Let $f_1(\epsilon') = \left(\frac{8SX^2V}{N} + \frac{4BX}{N}\right)\epsilon' + \frac{4SX}{N}(4SXV + 3B)$ be an affine function of $\epsilon'$, we can bound the gradient difference w.r.t $W_V$ as

$$\|\nabla_{W_V}\mathcal{L}_i(\phi_{\text{target}}^t) - \nabla_{W_V}\mathcal{L}_j(\phi_{\text{target}}^t)\|_F \leq f_1(\epsilon') \tag{22}$$

**Bound for $W_Q$.** We simplify the second term in Equation 18 as follows:

$$\|\nabla_{W_Q}\mathcal{L}_i(\phi_{\text{target}}^t) - \nabla_{W_Q}\mathcal{L}_j(\phi_{\text{target}}^t)\|_F^2$$

$$\stackrel{(i)}{=} \frac{2}{ND}\left\|(S_iX_iW_V - y_i)(I_N \otimes W_V^T X_i^T)\frac{\partial S_i}{\partial A_i}\frac{X_i \otimes X_iW_K}{\sqrt{D}} - (S_jX_jW_V - y_j)(I_N \otimes W_V^T X_j^T)\frac{\partial S_j}{\partial A_j}\frac{X_j \otimes X_jW_K}{\sqrt{D}}\right\|_F$$

$$\stackrel{(ii)}{=} \frac{2}{ND}\left\|(S_iX_iW_V - y_i)(I_N \otimes W_V^T X_i^T)\frac{\partial S_i}{\partial A_i}\left(\frac{X_i \otimes X_iW_K - X_j \otimes X_jW_K}{\sqrt{D}}\right)\right.$$

$$\left. + \left((S_iX_iW_V - y_i)(I_N \otimes W_V^T X_i^T)\frac{\partial S_i}{\partial A_i} - (S_jX_jW_V - y_j)(I_N \otimes W_V^T X_j^T)\frac{\partial S_j}{\partial A_j}\right)\frac{X_j \otimes X_jW_K}{\sqrt{D}}\right\|_F$$

$$= \frac{2}{ND}\left\|(S_iX_iW_V - y_i)(I_N \otimes W_V^T X_i^T)\frac{\partial S_i}{\partial A_i}\left(\frac{X_i \otimes (X_i - X_j)W_K + (X_i - X_j) \otimes X_jW_K}{\sqrt{D}}\right)\right.$$

$$+ \left[(S_iX_iW_V - y_i)\left((I_N \otimes W_V^T X_i^T)\left(\frac{\partial S_i}{\partial A_i} - \frac{\partial S_j}{\partial A_j}\right) + (I_N \otimes W_V^T(X_i^T - X_j^T))\frac{\partial S_j}{\partial A_j}\right)\right.$$

$$\left.\left. + \left\{\left(S_i(X_i - X_j) + (S_i - S_j)X_j\right)W_V + (y_j - y_i)\right\}(I_N \otimes W_V^T X_j^T)\frac{\partial S_j}{\partial A_j}\right]\frac{X_j \otimes X_jW_K}{\sqrt{D}}\right\|_F$$

$$\stackrel{(iii)}{\leq} \frac{8\sqrt{N}VX^3K}{D^{3/2}}\|F_i - y_i\|_F + \frac{2\sqrt{N}SX^4V^2K}{D^{3/2}} + \frac{\sqrt{N}V^2X^4K}{D^{3/2}}\|S_i - S_j\|_F + \frac{2B\sqrt{N}VX^3K}{D^{3/2}}$$

$$= \frac{\sqrt{N}V^2X^4K}{D^{3/2}}\|S_i - S_j\|_F + \frac{2\sqrt{N}VX^3K}{D^{3/2}}\left[4\alpha + SVX + B\right] \tag{23}$$

where (i) from Lemma 1; (ii) rearranging; (iii) uses triangle inequality and applies the assumptions on the bounds.

Plugging Equation 20 into Equation 23, we have

$$
\|\nabla_{W_Q}\mathcal{L}_i(\phi_{\text{target}}^t) - \nabla_{W_Q}\mathcal{L}_j(\phi_{\text{target}}^t)\|_F \leq \frac{2\sqrt{N}V^2X^4K}{D^{3/2}}\|\chi_i - \chi_j\|_F + \frac{2\sqrt{N}VX^3K}{D^{3/2}}\Big[4\alpha + 2SVX + B\Big]
$$
$$
= \frac{2\sqrt{N}V^2X^4K}{D^{3/2}}\cdot\epsilon' + \frac{2\sqrt{N}VX^3K}{D^{3/2}}\Big[4\alpha + 2SVX + B\Big] \quad (24)
$$

Let $f_2(\epsilon') = \frac{2\sqrt{N}V^2X^4K}{D^{3/2}}\cdot\epsilon' + \frac{2\sqrt{N}VX^3K}{D^{3/2}}\Big[4\alpha + 2SVX + B\Big]$ be an affine function of $\epsilon'$, we can bound the gradient difference w.r.t $W_Q$ as

$$
\|\nabla_{W_Q}\mathcal{L}_i(\phi_{\text{target}}^t) - \nabla_{W_Q}\mathcal{L}_j(\phi_{\text{target}}^t)\|_F \leq f_2(\epsilon') \quad (25)
$$

**Bound for $W_K$.** Similarly, we have the following bound for the third term

$$
\|\nabla_{W_K}\mathcal{L}_i(\phi_{\text{target}}^t) - \nabla_{W_K}\mathcal{L}_j(\phi_{\text{target}}^t)\|_F \leq f_3(\epsilon') \quad (26)
$$

where $f_3(\epsilon') = \frac{2\sqrt{N}V^2X^4Q}{D^{3/2}}\cdot\epsilon' + \frac{2\sqrt{N}VX^3Q}{D^{3/2}}\Big[4\alpha + 2SVX + B\Big]$ be affine function of $\epsilon'$.

Plugging all these expressions in 18, we have the following expression for the upper bound of the target-model gradient difference

$$
\|\nabla\mathcal{L}_i(\phi_{\text{target}}^t) - \nabla\mathcal{L}_j(\phi_{\text{target}}^t)\|_F \leq \sqrt{f_1(\epsilon')^2 + f_2(\epsilon')^2 + f_3(\epsilon')^2} \quad (27)
$$

Using Cauchy-Schwarz,

$$
\|\nabla\mathcal{L}_i(\phi_{\text{target}}^t) - \nabla\mathcal{L}_j(\phi_{\text{target}}^t)\|_F \leq \sqrt{f_1(\epsilon')^2 + f_2(\epsilon')^2 + f_3(\epsilon')^2}
$$
$$
\leq |f_1(\epsilon')| + |f_2(\epsilon')| + |f_3(\epsilon')|
$$
$$
= \underbrace{\left[\frac{2\sqrt{N}V^2X^4(K+Q)}{D^{3/2}} + \left(\frac{8SX^2V}{N} + \frac{4BX}{N}\right)\right]}_{c_1}\cdot\epsilon'
$$
$$
+ \underbrace{\frac{2\sqrt{N}VX^3}{D^{3/2}}\Big[(4\alpha + 2SVX + B)(Q+K)\Big] + \frac{4SX}{N}(4SXV + 3B)}_{c_2}
$$

Assume the model weights $Q$, $K$, and $V$ are bounded by $\frac{c}{\sqrt{3}}$, so that $\|\phi_{\text{target}}^t\| \leq c$. We further assume the ground-truth embedding $B$ and loss $\alpha$ are bounded. Such boundedness assumptions are standard in theoretical analyses of Transformers, including generalization and stability (Li et al., 2023b), as well as gradient-based optimization dynamics (Song et al., 2024). In practice, they are enforced by weight decay, and layer/activation normalization (Xiong et al., 2020).

Since $X$ passes through an RMS normalization layer with gain $g$, we have $X \leq g\sqrt{ND}$. Under this setting, the constants $c_1$ and $c_2$ simplify to:

$$c_1 = \mathcal{O}\left(\tfrac{4}{\sqrt{3}} N^2 \sqrt{ND}\, c^3 g^4\right), \tag{28}$$

$$c_2 = \mathcal{O}\left(\tfrac{8}{3\sqrt{3}} N^3 \sqrt{D}\, c^2 g^4\right). \tag{29}$$

If the RMS gain is sufficiently small, i.e.,

$$g < N^{-5/8} D^{-1/8} c^{-3/4} \quad \Leftrightarrow \quad g^4 < N^{-5/2} D^{-1/2} c^{-3},$$

then the bounds on $c_1$ and $c_2$ reduce to:

$$c_1 \leq \tfrac{4}{\sqrt{3}}, \tag{30}$$

$$c_2 \leq \tfrac{8\sqrt{N}}{3\sqrt{3}c}. \tag{31}$$

Therefore, the gradient distance can be bounded as

$$\|\nabla\mathcal{L}_i(\phi_{\text{target}}^t) - \nabla\mathcal{L}_j(\phi_{\text{target}}^t)\|_F \leq \tfrac{4}{\sqrt{3}} \cdot \epsilon' + \tfrac{8\sqrt{N}}{3\sqrt{3}c}.$$

$\square$

### A.2 PROOF OF THEOREM 4.2

We first have the following lemma.

**Lemma 3** ((Zhang, 2023), Theorem 1.8). *Let $f\colon \mathbb{R}^n \to \mathbb{R}$ be a twice continuously differentiable function. Given $x, y \in \mathbb{R}^n$, for every $z = x + t(y - x)$ where $t \in [0, 1]$, there exists a constant, $\zeta = \sup\|\nabla^2 f(x)\|$, such that*

$$\|\nabla f(z) - \nabla f(x)\|_F \leq \zeta\|z - x\|_F \tag{32}$$

*Proof.*

$$\|\nabla\mathcal{L}_i(\phi_{\text{target}}^{t_z}) - \nabla\mathcal{L}_j(\phi_{\text{target}}^{t_z})\|_F \tag{33}$$

$$\overset{(i)}{\leq} \|\nabla\mathcal{L}_i(\phi_{\text{target}}^{t_z}) - \nabla\mathcal{L}_i(\phi_{\text{target}}^{t_1})\|_F + \|\nabla\mathcal{L}_j(\phi_{\text{target}}^{t_z}) - \nabla\mathcal{L}_j(\phi_{\text{target}}^{t_1})\|_F + \|\nabla\mathcal{L}_i(\phi_{\text{target}}^{t_1}) - \nabla\mathcal{L}_j(\phi_{\text{target}}^{t_1})\|_F \tag{34}$$

$$\overset{(ii)}{\leq} 2\beta\|\phi_{\text{target}}^{t_z} - \phi_{\text{target}}^{t_1}\| + \Delta_{ij}^{t_1} \tag{35}$$

$$\overset{(iii)}{\leq} 2\delta\beta + \Delta_{ij}^{t_1} \tag{36}$$

where (i) from triangle inequality; (ii) from Lemma 3; (iii) from bounded curvature assumption and the definition of $\delta$. Similarly, we have

$$\|\nabla\mathcal{L}_i(\phi_{\text{target}}^{t_z}) - \nabla\mathcal{L}_j(\phi_{\text{target}}^{t_z})\|_F \leq 2\delta\beta + \Delta_{ij}^{t_2} \tag{37}$$

Combining these two above inequalities, we have

$$\|\nabla\mathcal{L}_i(\phi_{\text{target}}^{t_z}) - \nabla\mathcal{L}_j(\phi_{\text{target}}^{t_z})\|_F \leq 2\delta\beta + \max\{\Delta_{ij}^{t_1}, \Delta_{ij}^{t_2}\} \tag{38}$$

$\square$

### A.3 Convergence of XMAS

**Corollary A.1** (Convergence of XMAS). *Under the assumptions of Theorem 4.2, applying incremental gradient methods with stepsize $\eta$ on subsets found by XMAS, converges to a neighborhood of the solution $\phi^*$ found by training on full data:*

$$\|\phi^{t+1} - \phi^*\|^2 \leq (1 - \eta c')^{t+1}\|\phi^t - \phi^*\|^2 + \frac{2\xi R'}{c'^2} + \eta B^2 \left(\frac{r_{\min}}{k}\right)^2 g_{\max}^2 \tag{39}$$

*where $c' \leq \|H\|$, $B$ is the total size of the subset, $g_{\max}$ is the largest gradient norm of individual examples during training, $r_{\min}, r_{\max}$, are the size of the smallest and largest clusters, $R' = \min\{d_0, Bg_{\max} + \xi/c'\}$ and $d_0 = \|\phi^0 - \phi^*\|$ is the initial distance to the optimal solution $\phi^*$, and $\xi = K[r_{\min}\Delta_2 + (r_{\max} - r_{\min})g_{\max}]$.*

Our proof is similar to the one in S2L (Yang et al., 2024, Appendix A.2). We provide the details in order to ensure completeness.

*Proof.* Without loss of generality, assume we select $k$ examples from each cluster and we have $k \leq \min_{j \in [K]} |C_j|$. Then the error of the subset in capturing the full gradient will be

$$\xi \leq \sum_j (|C_j| - k)(\bar{g}_j + \Delta), \tag{40}$$

where $\bar{g}_j$ is the norm of the average gradient of all samples from $C_j$. Here $\Delta$ is the maximum error in gradients between two different samples.

In practice, we can weight elements of the subset by $r_{\min}/k$, which has a similar effect to scaling the step size when training on the subset. Let $g_{\max} = \max_j \|g_j\|$ be the maximum gradient norm during training, $r_{\max} = \max_j |C_j|$, $r_{\min} = \min_j |C_j|$. Then, we get

$$\xi' \leq \sum_j [(r_{\min} - k)\Delta + (|C_j| - r_{\min})(\bar{g}_j + \Delta)] \tag{41}$$

$$\leq K[r_{\min}\Delta + (r_{\max} - r_{\min})g_{\max}]. \tag{42}$$

The first term in the RHS of Eq. (7) is the error of the subset selected from $C_j$ to capture its full gradient and the second term is due to selecting the same number of examples, $k$, from the larger clusters. Using the above error and following the proof of Theorem 1 in Mirzasoleiman et al. (2020), for a constant step size $\alpha \leq 1/c$ we get:

$$\|\theta^{t+1} - \theta^*\|^2 \leq (1 - \alpha c)^{t+1}\|\theta^t - \theta^*\|^2 + \frac{2\xi' R}{c^2} + \alpha B^2 \left(\frac{r_{\min}}{k}\right)^2 g_{\max}^2, \tag{43}$$

where $c \leq \|H\|$, and $B = k \cdot K$ is the total size of the subset, $R = \min\{d_0, Bg_{\max} + \xi'/c\}$ and $d_0 = \|\theta^0 - \theta^*\|$ is the initial distance to the optimal solution $\theta^*$.

If $k \geq |C_j|$ for any cluster $C_j$, one can simply add $(r_{\min}/k - 1) \cdot \hat{g}_j$ to $\xi'$ for the corresponding clusters, where $\hat{g}_j$ is the norm of the total gradient of cluster $C_j$ and we replace $r_{\min}$ in Eq. (7) with the size of smallest cluster that has larger than $k$ examples. $\square$

## A.4 Proofs of Lemmas

**Proof of Lemma 1.**

*Proof.*     1. Applying chain rule and simplifying, we get

$$\nabla_{W_V}\mathcal{L}_i(\phi^t) = \nabla_{F_i}\mathcal{L}_i(\phi^t) \cdot \nabla_{W_V} F_i(\phi^t)$$
$$= (F_i - Y_i)(S_i X_i \otimes I_D)$$

2. Applying chain rule and simplifying, we get

$$\nabla_{W_Q}\mathcal{L}_i(\phi^t) = \nabla_{F_i}\mathcal{L}_i(\phi^t) \cdot \nabla_{S_i} F_i(\phi^t) \cdot \nabla_{A_i} S_i(\phi^t) \cdot \nabla_{W_Q} A_i(\phi^t)$$
$$= (F_i - Y_i)(I_N \otimes W_V^T X_i^T) \cdot \nabla_{A_i} S_i(\phi^t) \cdot \left(\frac{X_i \otimes X_i W_K}{\sqrt{D}}\right)$$

3. Proceeding as above for $\nabla_{W_Q}\mathcal{L}_i(\phi^t)$, we get

$$\nabla_{W_K}\mathcal{L}_i(\phi^t) = (S_i X_i W_V - y_i)(I_N \otimes W_V^T X_i^T)\frac{\partial S_i}{\partial A_i}\left(\frac{X_i \otimes X_i W_Q}{\sqrt{D}}\right)\Lambda_{d,D}$$

where $\Lambda_{d,D}$ is the commutation matrix.

$\square$

**Proof of Lemma 2.**

*Proof.* For a single sample, $i$, we have
$$S_i = softmax(A_i)$$
where softmax is applied row-wise to A. The expression for a single element in $S_i$ present at (j, k) is given as:
$$(S_i)_{jk} = \frac{e^{(A_i)_{jk}}}{\sum_{z=1}^{L} e^{(A_i)_{jz}}}$$

Now, clearly, we have
$$\frac{\partial(S_i)_{jk}}{\partial(A_i)_{mn}} = 0 \ \text{ if } j \neq m$$

Now for a specific row, $i$, the Jacobian of the $i$-th row of $S_i$ wrt the $i$-th row of $A_i$ is given as

$$J_{kn}^{(j)} = \frac{\partial(S_i)_{jk}}{\partial(A_i)_{jn}} = \begin{cases} (S_i)_{jk}\Big(1 - (S_i)_{jk}\Big) & \text{if } k = n \\ -(S_i)_{jk}(S_i)_{jn} & \text{if } k \neq n \end{cases}$$
$$= (S_i)_{jk}\Big(\delta_{kn} - (S_i)_{kn}\Big)$$

where $\delta_{jm}$ is the Kronecker delta (1 if j=m, 0 otherwise).

Now the complete Jacobian of the attention matrix can be written as follows:

$$(\nabla_{A_i}S_i)_{jkmn} = \delta_{jm}(S_i)_{jk}\Big(\delta_{kn} - (S_i)_{kn}\Big)$$

where $\delta_{jm}$ and $\delta_{kn}$ are the Kronecker deltas.

Following is the expression for the Frobenius Norm of this Jacobian:

$$\|(\nabla_{A_i} S_i)_{jkmn}\|_F^2 = \sum_{j=1}^{N} \sum_{k=1}^{N} \sum_{m=1}^{N} \sum_{n=1}^{N} \left( \frac{\partial (S_i)_{jk}}{\partial (A_i)_{mn}} \right)^2$$

$$= \sum_{j=1}^{N} \sum_{m=1}^{N} \sum_{n=1}^{N} \left( \frac{\partial (S_i)_{jk}}{\partial (A_i)_{jn}} \right)^2$$

$$\|(\nabla_{A_i} S_i)_{jkmn}\|_F^2 = \sum_{j=1}^{N} \|J^{(j)}\|_F^2 \tag{44}$$

Now computing $\|J^{(j)}\|_F$ for a single row $j$. Also, let $\xi_p^{(j)} = \sum_{q=1}^{N} (S_i)_{jq}^p$ Then we have

$$\|J^{(j)}\|_F^2 = \sum_{k=1}^{N} \sum_{n=1}^{N} \left( J_{kn}^{(j)} \right)$$

$$= \sum_{k=1}^{N} \left( (S_i)_{jk}(1 - (S_i)_{jk}) \right)^2 + \sum_{k=1}^{N} \sum_{n \neq k} \left( -(S_i)_{jk}(S_i)_{jn} \right)^2$$

$$= \xi_2^{(j)} - 2\xi_3^{(j)} + \left( \xi_2^{(j)} \right)^2$$

Now the maximum value of $\|J^{(j)}\|_F^2$ under constraints, $\xi_1^{(j)} = 1$ and $(S_i)_{jq} \geq 0 \;\; \forall q \in \{0, 1, 2, \dots, N\}$, is $\frac{1}{4}$ and is achieved when exactly two $q$, $(S_i)_{jq} = \frac{1}{2}$ when $N \geq 2$. Plugging this in equation 44, we get

$$\|(\nabla_{A_i} S_i)_{jkmn}\|_F^2 = \sum_{j=1}^{N} \|J^{(j)}\|_F^2 \leq \sum_{j=1}^{N} \left( \frac{1}{4} \right) = \frac{N}{4}$$

Therefore, we have

$$\|(\nabla_{A_i} S_i)_{jkmn}\|_F \leq \frac{\sqrt{N}}{2}$$

$\square$

## B  ADDITIONAL EXPERIMENTAL SETTINGS

**Models.** For the target LVLMs, we use the pre-trained LLaVA-1.5-7B, LLaVA-1.5-13B models Liu et al. (2024a), LLaVA-1.6-Mistral-7B Li et al. (2024) and Phi-3.5-Vision-Instruct‘Abdin et al. (2024). For the proxy models, we use the TinyLLaVA (Zhou et al., 2024) with 2 different scales 0.5B and 2.0B. The default one is TinyLLaVA 2.0B due to its superior performance. Table 2 summarizes the language model and vision encoder of different models used in our experiment.

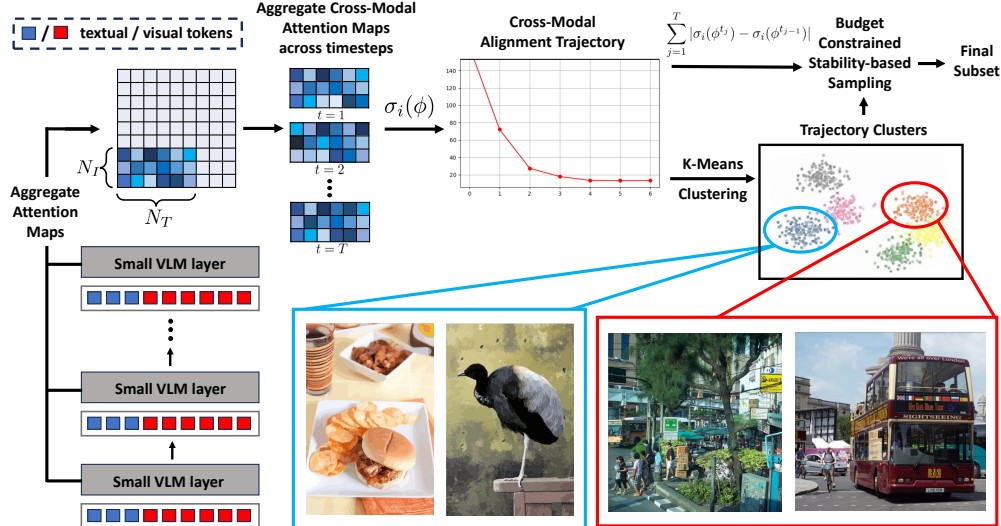

Figure 5: XMAS employs a small proxy VLM to find alignment trajectory for examples in the fine-tuning data. Examples with similar alignment trajectory have similar gradients during instruction tuning. Then, it clusters the alignment trajectories and sample a balanced subset of examples with more stable trajectories from the clusters.

Table 2: Details of the model architectures used in our experiments are provided below. Model names correspond to their repository names on HuggingFace.

| Model | Language model | Vision encoder |
|---|---|---|
| LLaVA-1.5-13B | lmsys/vicuna-13b-v1.5 | openai/clip-vit-large-patch14 |
| LLaVA-1.5-7B | lmsys/vicuna-7b-v1.5 | openai/clip-vit-large-patch14 |
| TinyLLaVA 2.0B | stablelm/stablelm-2-zerphyr-1_6b | openai/clip-vit-large-patch14 |
| TinyLLaVA 0.5B | Qwen/Qwen2-0.5B | google/siglip-so400m-patch14-384 |

**VIT datasets.** We apply coreset selection to two distinct vision instruction tuning (VIT) datasets: LLaVA 665k Liu et al. (2024a) and Vision-Flan Xu et al. (2024), both of which are widely used benchmarks for evaluating multimodal instruction-following models. LLaVA 665k comprises approximately 665,000 VIT examples aggregated from 12 diverse vision-language datasets. These datasets span a wide range of tasks, such as image captioning, visual question answering, and visual reasoning, providing a comprehensive training mixture for instruction-tuned large vision-language models. The examples are automatically aligned and instruction-formatted, making the dataset suitable for large-scale fine-tuning. In contrast, Vision-Flan is a more task-structured benchmark composed of 191 individual vision-language tasks. Each task includes around 1,000 high-quality, expert-labeled VIT examples, leading to a total of about 186,000 samples. Unlike LLaVA 665k, which merges data across tasks, Vision-Flan preserves a task-level granularity, allowing for more fine-grained evaluation and task-specific data selection strategies.

**Training details.** In all experiments, we fine-tune the target models using LoRA Hu et al. (2022) for one epoch regardless of the subset size. We strictly follow the official finetuning hyperparameters specified in LLaVA-1.5. For Phi-3.5-Vision-Instruct, we used the settings in https://github.com/microsoft/PhiCookBook. For proxy models, we train full model (i.e., without LoRA) for one epoch, following the official hyperparameters specified in TinyLLaVA. This results in a total of $T = 7$ checkpoints for the

Table 3: A comparative analysis of the average relative performance of LLaVA-1.5-7B using different methods at different sampling ratios on the LLaVA 665k and Vision-Flan datasets.

| Method | LLaVA-665K | | | | Vision-Flan | | |
|---|---|---|---|---|---|---|---|
| | 10% | 20% | 30% | 50% | 5% | 10% | 15% |
| Rand | 93.5 | 96.3 | 97.4 | 99.2 | 92.7 | 96.2 | 98.4 |
| MP (Marion et al., 2023) | 92.4 | 95.4 | 97.9 | 99.6 | 92.9 | 95.6 | 97.7 |
| HL (Zhou et al., 2023b) | 90.2 | 95.4 | 96.6 | 99.8 | 91.4 | 95.8 | 97.5 |
| EL2N (Paul et al., 2021) | 93.3 | 95.0 | 95.6 | 99.3 | 88.7 | 91.9 | 87.4 |
| D2 Pruning (Maharana et al., 2023) | 91.8 | 94.1 | 95.5 | 97.7 | 92.4 | 97.6 | 99.0 |
| SemDeDup (Abbas et al., 2023) | 91.4 | 95.9 | 95.4 | 96.8 | 92.6 | 93.0 | 96.3 |
| CLIP-Score (Hessel et al., 2021) | 86.1 | 90.8 | 92.3 | 96.2 | 93.1 | 96.6 | 98.1 |
| Self-Sup (Sorscher et al., 2022) | 85.9 | 91.6 | 94.6 | 97.8 | 85.7 | 92.4 | 99.1 |
| Self-Filter (Chen et al., 2024) | 89.3 | 88.4 | 93.7 | 97.7 | 90.3 | 93.6 | 94.5 |
| COINCIDE (Lee et al., 2024) | 94.1 | 96.6 | 98.7 | 98.8 | 92.5 | 97.0 | 98.3 |
| XMAS (**Ours**) | **95.4** | **97.1** | **99.2** | **100.0** | **96.1** | **98.9** | **100.2** |

trajectory. For distributed training, we use DeepSpeed Rasley et al. (2020). For K-means, we set the number of clusters $K$ to 1000 and use the GPU version of the faiss library Johnson et al. (2019).

**Evaluation datasets.** We evaluate the performance of the fine-tuned target models across four core capabilities: reasoning, hallucination resistance, visual perception, and cognition. To comprehensively assess these abilities, we utilize a diverse suite of both academic-task-oriented benchmarks and recent benchmarks tailored for instruction-following LMMs, totaling ten evaluation datasets.

For visual perception and cognition, we include VQAv2Goyal et al. (2017) and GQAHudson & Manning (2019), which require open-ended answers to visual questions, assessing the model's ability to understand and interpret images. VizWizGurari et al. (2018), a dataset comprising real-world images taken by visually impaired users, is used to test the model's zero-shot generalization capabilities in a more challenging, accessibility-focused setting. TextVQASingh et al. (2019) measures performance on text-rich visual inputs, challenging models to combine OCR with multimodal reasoning. For science-focused reasoning, we adopt the image subset of ScienceQA Lu et al. (2022), which consists of multiple-choice scientific questions accompanied by relevant visual content.

To evaluate hallucination behavior, we employ POPE Li et al. (2023a), which measures the model's tendency to generate factually incorrect information in multimodal contexts. POPE includes three subsets—random, common, and adversarial samples from the COCO dataset and we report the average F1 score across all splits.

For general reasoning and robustness, we use several recently proposed benchmarks. MME-PerceptionLiang et al. (2024) tests perception using binary (yes/no) questions based on visual content, while MMBenchLiu et al. (2024b) evaluates the robustness of multiple-choice answers across a broad range of tasks. MMVetYu et al. (2023) and LLaVABenchLiu et al. (2023) focus on visual conversation abilities, evaluating both the correctness and helpfulness of model responses using GPT-4 as a judge.

By covering a wide spectrum of domains, ranging from scientific reasoning and accessibility to free-form visual conversations, this evaluation protocol provides a comprehensive measure of how well the fine-tuned models generalize across real-world and task-specific scenarios.

**Evaluation metric.** We follow the same evaluation protocols outlined in LLaVA-1.5. Similar to COINCIDE, we measure the relative performance as (model performance / full-finetuned performance) × 100% to assess

Table 4: Average relative performances (ARP) over full data when training LLaVa-1.5-7B on 10% subsets of LLaVA 665K found by XMAS when using different (a) alignment score, (b) matrix aggegation, and (c) instability score.

| Score | ARP |
|---|---|
| Sum values | **95.4** |
| Concat values | 93.0 |
| Sing vector | 93.4 |

(a) Alignment score

| Method | ARP |
|---|---|
| Sum | **95.4** |
| Concat text | 92.9 |
| Concat vision | 92.1 |

(b) Matrix aggregation

| Score | ARP |
|---|---|
| Sum abs diff | **95.4** |
| Sum sq diff | 94.8 |
| Variance | 93.9 |

(c) Instability score

Table 5: Average relative performances (ARP) over full data when training LLaVa-1.5-7B on 10% subsets of LLaVA 665K found by XMAS when using different (a) cluster sampling strategies, (b) attention matrices, and (c) found by COINCIDE with different layer choices.

| Attn layer | ARP |
|---|---|
| First layer | 92.9 |
| Last layer | 93.7 |
| All layers | **95.4** |

(a) Attention layer

| Singular values | ARP |
|---|---|
| All | **95.6** |
| Top-5 | 95.4 |
| Top-1 | 94.2 |

(b) Singular values

| Layer indices | ARP |
|---|---|
| 3,7,11,15,19 | **94.1** |
| 4,8,12,16,20 | 91.6 |
| 3,8,13,18,23 | 92.1 |

(c) Layer choice

the performance of subsets compared to full dataset. To compare between different methods, we **A**verage the **R**elative **P**erformance (ARP) across all evaluation datasets.

**Computational resources.** All experiments are conducted using 8 NVIDIA RTX A6000 GPUs.

## C ADDITIONAL EXPERIMENTAL RESULTS

**Quantitative results of LLaVA 665k and Vision-Flan.** In this section, we present the detailed Average Relative Performance (ARP) corresponding to the experiments in Section 5.1. The full ARP results underlying Figures 1 are reported in Table 3. As shown, our method consistently achieves the highest performance across various subset budgets on both LLaVA 665k and Vision-Flan datasets. Notably, XMAS is the only approach that consistently outperforms random sampling on LLaVA 665k. On Vision-Flan, XMAS outperforms COINCIDE by nearly 2% when selecting subsets in the 5–15% range.

**Choice of alignment score.** Table 4a compares different choices of the alignment score (Definition 4.1). For a single checkpoint, instead of summing the top-5 singular values, one option is to concatenate them into a 5-dimensional vector, resulting in an alignment trajectory of size 35 across 7 checkpoints. Another option is to use the singular vector (dimension 576) corresponding to the largest singular value, giving a trajectory of size 4032 in Equation 7. As shown, summing the top-5 singular values outperforms the alternatives by about 2%.

**Choice of layer aggregation strategies.** In Definition 4.1, we aggregate the cross-modal attention matrices by summing them across all layers, resulting in a single matrix of size $n_T \times n_I$. In this experiment, we benchmark this approach with alternative strategies that stack the attention matrices along either the text or vision dimension, producing matrices of size $Ln_T \times n_I$ or $n_T \times Ln_I$, respectively, where $L$ is the number of layers. As shown in Table 4b, summing the per-layer attention matrices outperforms both stacking approaches. We hypothesize that summation reduces noise and highlights shared structures across layers,

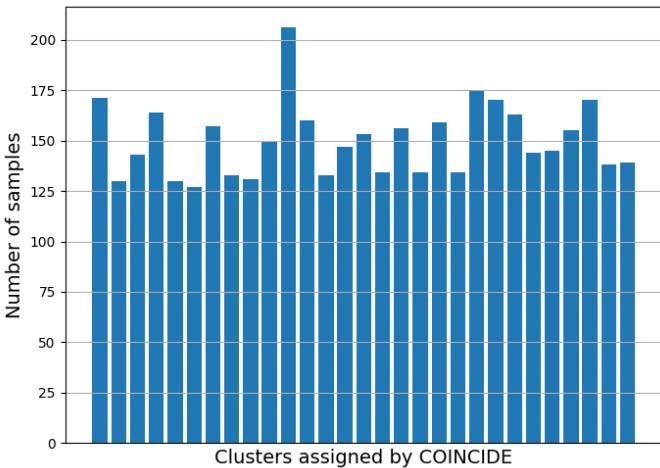

Figure 6: Distribution of concepts, i.e., clusters found by COINCIDE on LLaVA 665k, in the largest cluster found by XMAS. Only concepts with more than 100 samples are shown in the figure.

Table 6: Average relative performances (ARP) comparison of different 10% subsets of the LLaVA 665k dataset across target models. The best method is shown in **bold**, and the second best is underlined. Results marked with * report ARP without VizWiz evaluation due to its unavailability on EvalAI.

| Method | LLaVA-1.5-7B | LLaVA-1.5-13B | LLaVA-1.6-Mistral-7B* | Phi-3-Vision-Instruct* |
|---|---|---|---|---|
| Rand | 93.5 | 93.7 | 91.9 | 101.8 |
| MP | 92.4 | 92.5 | 88.8 | 101.5 |
| HL | 90.2 | 93.7 | 78.1 | 102.2 |
| EL2N | 93.3 | 93.3 | 87.8 | 101.7 |
| D2 Pruning | 91.8 | 91.8 | 89.5 | 101.7 |
| SemDeDup | 91.4 | 90.8 | 92.1 | 102.6 |
| CLIP-Score | 86.1 | 89.9 | 86.7 | 102.3 |
| Self-Sup | 85.9 | 90.2 | 88.9 | 102.1 |
| Self-Filter | 89.3 | 89.8 | 88.4 | 102.4 |
| COINCIDE | 94.1 | 94.6 | 91.6 | 101.9 |
| XMAS (**Ours**) | **95.4** | **95.6** | **93.8** | **103.3** |

leading to more robust alignment signals. Moreover, this strategy is computationally more efficient, as computing the SVD on the summed matrix is significantly faster than on the higher-dimensional stacked variants. Therefore, we adopt layer-wise summation as our default aggregation method.

**Choice of instability score.** Table 4c compares alternative definitions of the instability score for alignment trajectories. Our default choice, the sum of absolute differences $S_i = \sum_{j=1}^{T} |\sigma_i(\phi^{t_j}) - \sigma_i(\phi^{t_{j-1}})|$, achieves the best performance (95.4 ARP) as it directly measures cumulative oscillation across checkpoints. Using squared differences $S_i^{\text{sqr}} = \sum_{j=1}^{T} (\sigma_i(\phi^{t_j}) - \sigma_i(\phi^{t_{j-1}}))^2$ slightly reduces performance, since large fluctuations are overweighted. Variance $S_i^{\text{var}} = \frac{1}{T} \sum_{j=1}^{T} (\sigma_i(\phi^{t_j}) - \frac{1}{T} \sum_{k=1}^{T} \sigma_i(\phi^{t_k}))^2$ performs worse, as it only captures the spread of values and ignores the temporal ordering of oscillations.

**Choice of attention layers.** Table 5a illustrates that aggregating cross-modal matrices across all the layers outperforms only using the first or last layer.

**Number of singular values.** While using all the singular values of the cross-modal attention can capture its full spectrum, it is more computationally more expensive. Indeed, using only top-5 singular values reduces the SVD computation from 51.7 (ms) to 1.77 (ms) on average. Furthermore, using top-5 singular values only harms the average relative performance by 0.2% as detailed in Table 5b while using top-1 singular values decreases the performance more significantly by 1.4%.

**COINCIDE is sensitive to the layer choice.** We would like to emphasize that COINCIDE's total time includes the cost of training a proxy model and finding layers that encode different concepts, but the cost for "finding layers" are omitted from the time we reported, as we directly used the layers specified in their paper. For finding the layers, COINCIDE needs to try different sets of layers (as much as $\binom{24}{5}$) and retrain the target model on the corresponding subsets. This makes COINCIDE considerably more expensive than our method (although we are not able to calculate its exact cost). We empirically showed that COINCIDE is sensitive to the layer choice. Table 5c compares the ARP of COINCIDE when varying the layer used to extract features. Clearly, the performance decreases significantly when using a different set of layers. Furthermore, COINCIDE requires substantially more memory: for each data point it stores a feature vector of size 20480, while XMAS only stores a vector of size 35.

**Cluster diversity.** As illustrated in Figure 6, the clusters identified by XMAS contain samples spanning multiple concepts detected by COINCIDE. This indicates that while samples within an XMAS cluster are conceptually diverse, they exhibit redundancy relative to each other in terms of training—an aspect that concept-based clustering methods like COINCIDE fails to capture. Figures 7-10 show the semantic diversity within a single XMAS cluster for different datasets. The values in the cross-model attention trajectory graphs also show that these clusters vary in the levels of cross-modal alignment.

**Other target models.** To assess the generalizability of XMAS, we train two additional target models—LLaVA-1.6-Mistral-7B and Phi-3.5-Vision-Instruct—using 10% subsets selected by the same proxy model (TinyLLaVA 2.0B), and compare XMAS against other methods. Table 6 shows that XMAS consistently achieves the best performance and is the only method that outperforms random selection across all target model architectures. In contrast, COINCIDE relies on hand-picked features, which fail to generalize to these new models.

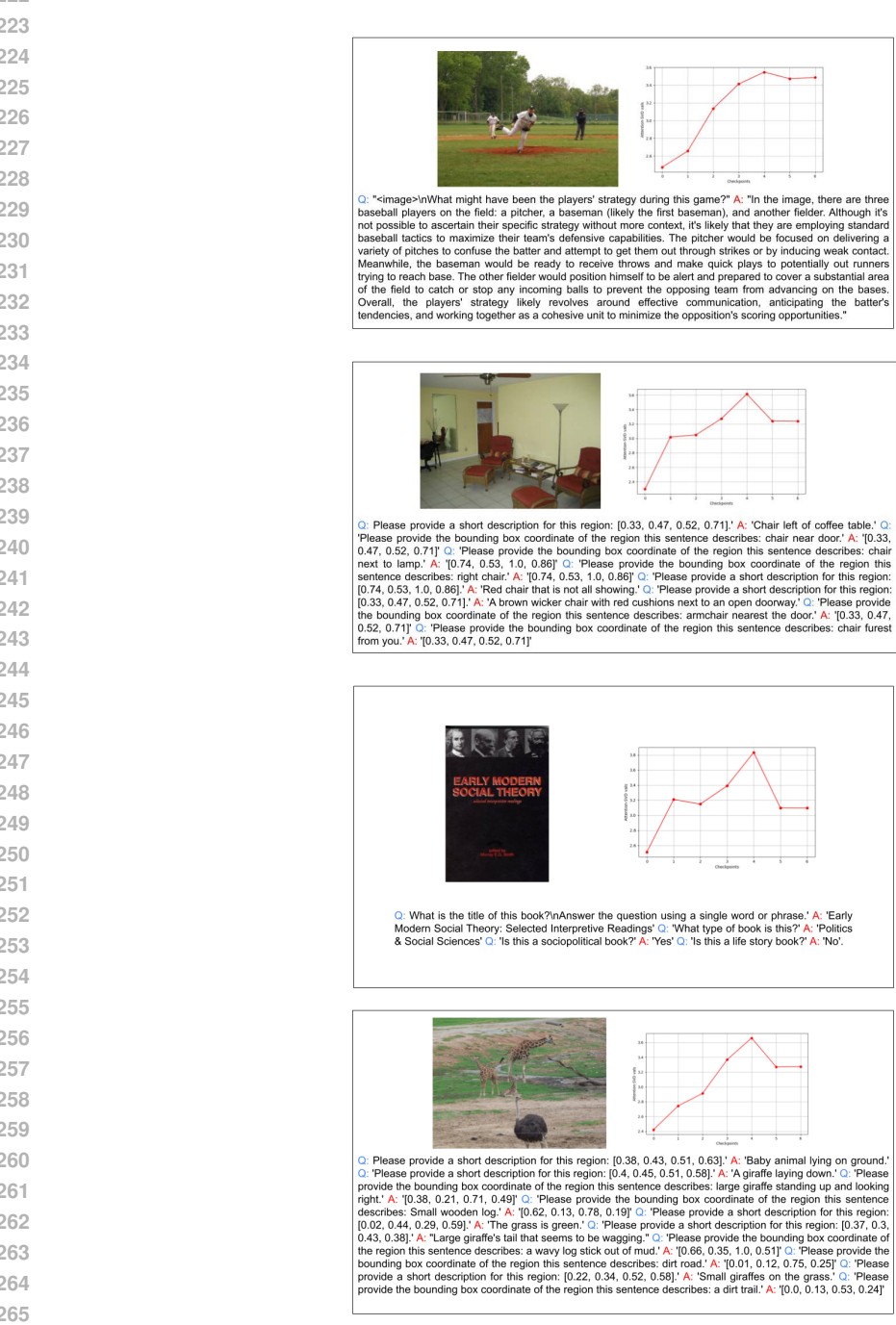

Figure 7: Samples from the largest XMAS clusters in LLaVA 665k dataset.

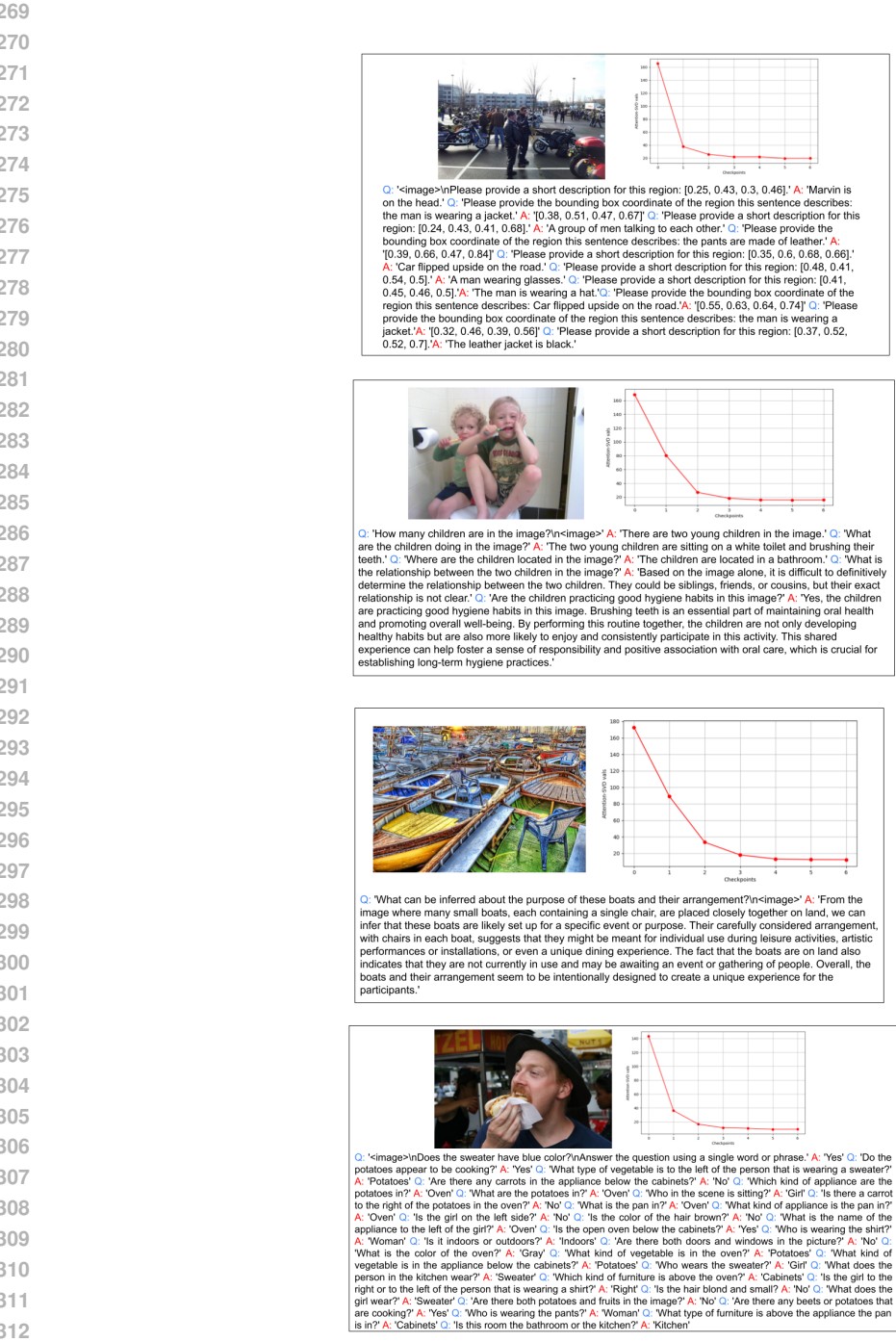

Figure 8: Samples from the smallest XMAS clusters in LLaVA 665k dataset.

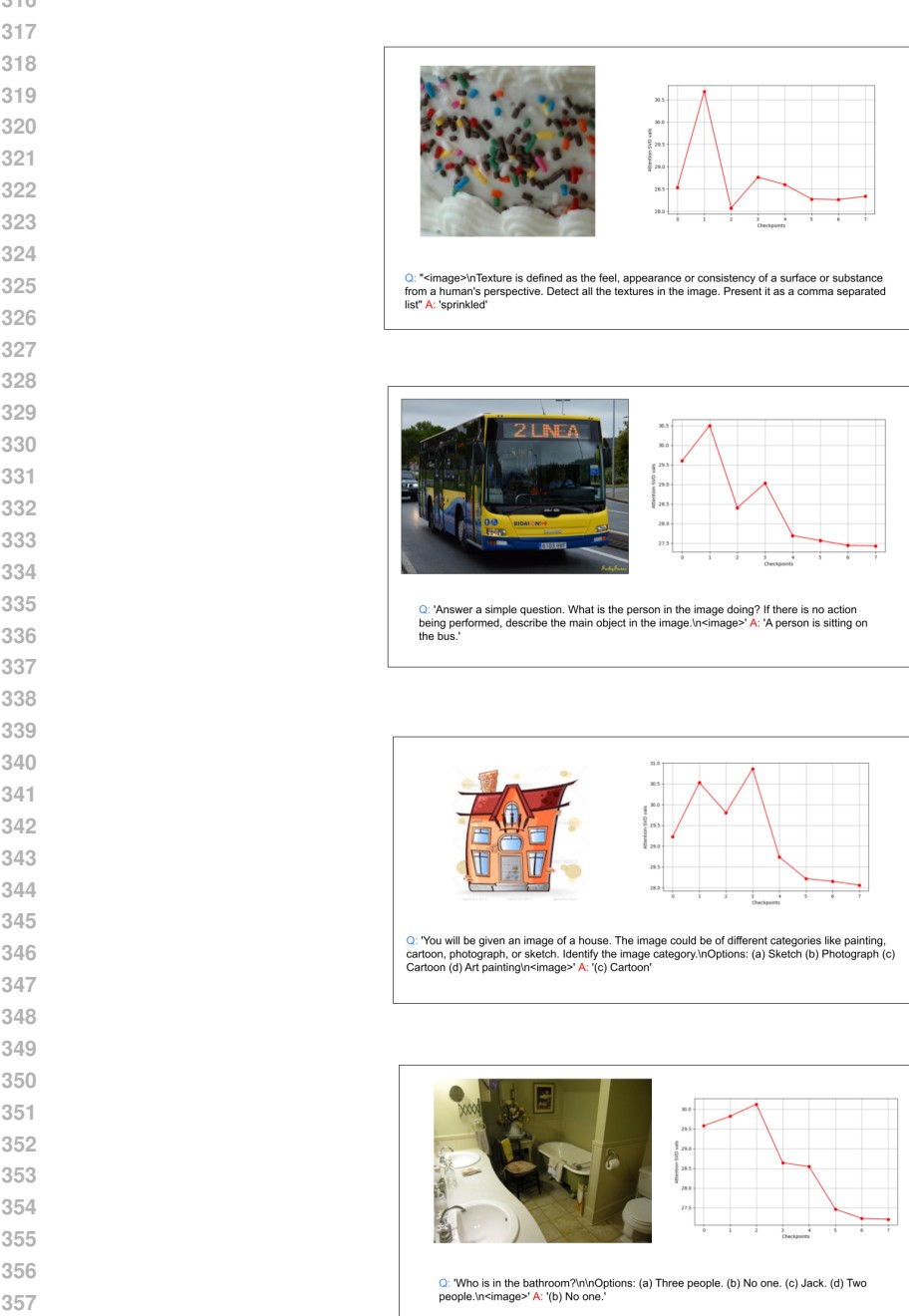

Figure 9: Samples from the largest XMAS clusters in Vision-Flan 191k dataset.

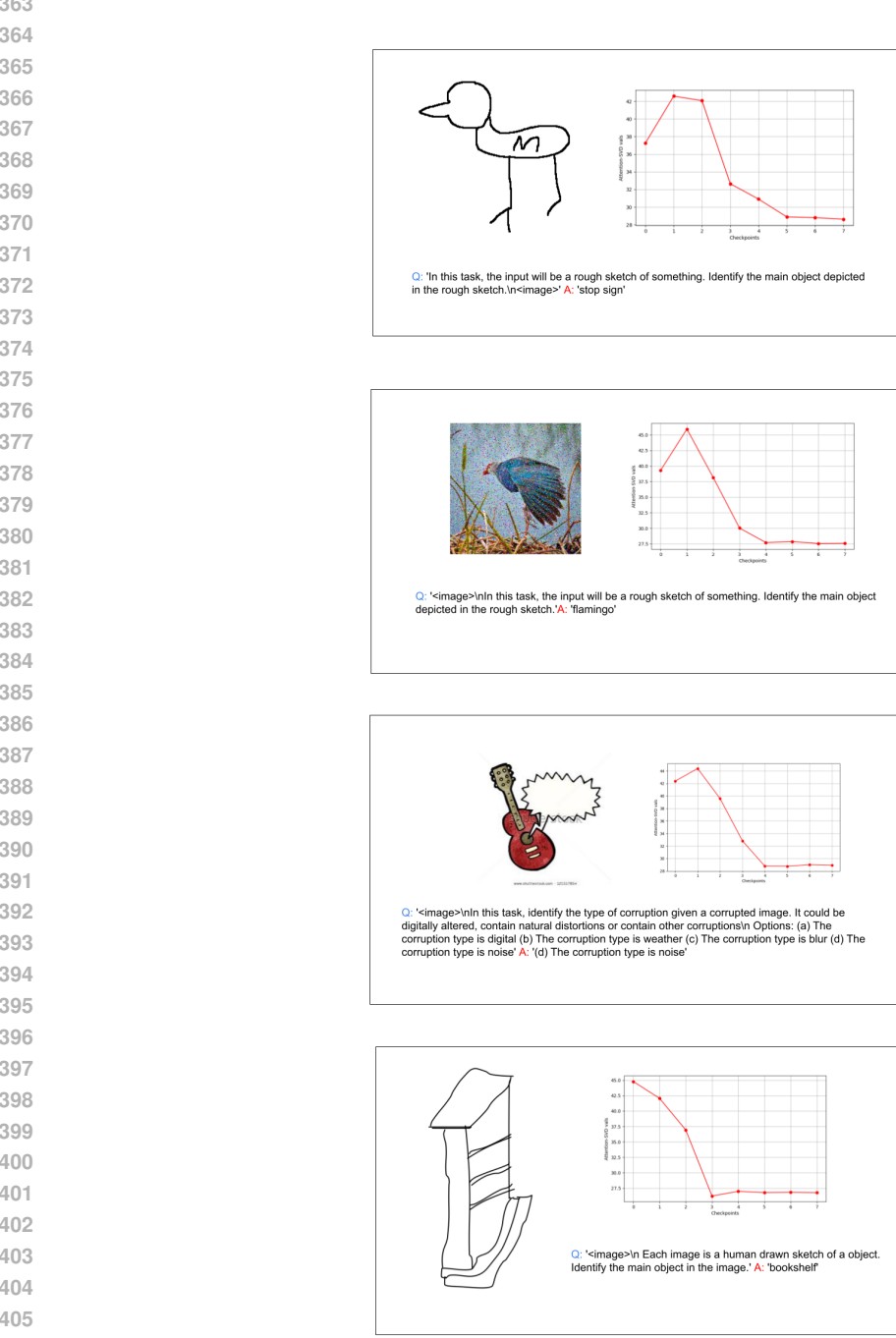

Figure 10: Samples from the smallest XMAS clusters in Vision-Flan 191k dataset.

