# OpenReview forum: "Data Selection for Fine-tuning Vision Language Models via Cross Modal Alignment Trajectories"
_ICLR.cc/2026/Conference — Submitted to ICLR 2026_

### Official Review · Reviewer_Vwim · 2025-10-26

**Soundness:** 3
**Presentation:** 3
**Contribution:** 3
**Rating:** 6
**Confidence:** 3

**Summary:**

This paper introduces XMAS, a framework for data selection in fine-tuning large vision-language models (LVLMs). The key idea is that examples with similar cross-modal attention matrices during instruction tuning also have similar gradients, and hence, are redundant for training. XMAS fine-tunes a small proxy LVLM to compute cross-modal alignment trajectories (the top singular values of attention matrices across training checkpoints), clusters examples with similar trajectories, and samples a balanced subset emphasizing stable trajectories.

**Strengths:**

**Theoretical Grounding:** The idea of clustering cross-modal alignment trajectories derived from a proxy LVLM is original and supported by theoretical justification.

**Strong Empirical Results:** Comprehensive experiments on two large datasets demonstrate clear improvements in data reduction efficiency (up to 30% better than the best baseline) with minimal performance loss.

**Weaknesses:**

**Incremental Technical Novelty:** While the theoretical analysis is elegant, the practical pipeline (proxy training + clustering + balanced sampling) remains conceptually close to COINCIDE (activation-based clustering). The novelty is incremental, largely reinterpreting feature-space redundancy through attention SVD trajectories.

**Dense Presentation:** The paper is well-organized but mathematically dense, with long proofs occupying many pages. Some key intuitions—such as why top singular values reflect gradient dynamics—could be better highlighted through figures or simplified derivations.

**Limited Validation:** The derivations (Theorems 4.1–4.2) make restrictive assumptions—single-layer transformers, small RMS gains, bounded curvature—that are not empirically verified in large-scale LVLMs. The link between these theoretical results and the practical proxy implementation is assumed rather than demonstrated.

**Questions:**

Please refer to the weakness section.

---

> ### Author Response · Authors · 2025-11-24
> **Response to Reviewer Vwim (part 1)**
>
> We thank the reviewer for recognizing the theoretical grounding and strong empirical results of our method. Below we would like to address the remaining concerns.
>
> ---
>
> **W1.** We do disagree with the reviewer about our novelty. Eliminating redundancy is the underlying idea for many data selection methods, including of our baselines COINCIDE, D2 Pruning, SemDeDup, CLIP Score, and SelfSup, and many of these methods use clustering or proxy models. However, as we discussed, all **these methods define redundancy in a heuristic manner and when applied to LVLMs, they often fail to outperform random selection**.
>
> Our work defines redundancy in a principled manner as “gradient similarity during the training” and is the first theoretically-rigorous method that outperforms random selection for selection subsets of varying sizes from different datasets. Below, we iterate the ideas behind COINCIDE and XMAS to clarify that **the main idea behind the two methods and the data points they select are entirely different**.
>
> **COINCIDE** is a purely heuristic method that aims to eliminate redundancy from dense parts of the data, based on activations of a *trained model (i.e., single checkpoint)*, and *does not balance the data or provide any guarantee*. To do so, it selects examples from the boundary of concept clusters that are found heuristically by clustering *activations of manually-selected layers* of a trained LVLM (COINCIDE is very sensitive to the choice of heuristic concepts/layers, c.f. Table 5c). As we discussed, similarity of examples at a single checkpoint does not imply that they affect the model parameters similarly “throughout the training” (i.e. redundancy). Hence, COINCIDE’s heuristic definition of redundancy does not work well in many settings. This is evident by the fact that **Random selection outperforms COINCIDE for selecting larger subsets**, i.e. >30% from LlaVa and >10% from VisionFlan.
>
> **XMAS** is a theoretically rigorous method that defines redundancy as *gradient similarity throughout the training* (not a single checkpoint), and *guarantees superior performance on any “unseen” downstream task*, by selecting a **(1) balanced and (2) non-redundant** subset of **central** examples from gradient clusters. To do so, XMAS bounds gradient differences via cross-modal attention differences and provides theoretical guarantees for the selected subset. This is evident by the fact that **XMAS outperforms random selection for selecting subsets of arbitrary sizes (small and large) from various datasets.**
>
> **Thus the two methods have different aims and select very different examples.**
>
> ---
>
> **W2**. As discussed above, XMAS aims to find clusters of examples with similar gradients and selects a balanced sample from these clusters. To do so, XMAS proves that attention distances upper bound gradient distances. Thus, it clusters examples based on the trajectory of their cross-modal attention matrices to find groups of examples with similar gradients. Top singular values of a matrix are closely related to its norm. So, examples that have similar top singular values for their cross-modal attention matrices have similar gradient norms (i.e. amount of cross-modal alignment). XMAS finds clusters of examples with similar gradient norms throughout the entire training. Within every cluster, examples are learned together (at a similar pace) and thus have similar learning dynamics. Every cluster contains subgroups of examples with similar gradient vectors (directions) with slightly different alignment trajectory patterns. By sampling examples with the most stable alignment trajectory, XMAS selects the central example from every gradient subgroup. Selecting a balanced subset of the most stable examples from alignment trajectory clusters ensures superior performance on any unseen downstream task.
>
> We have added the above clarification to our revision.

---

> > ### Author Response · Authors · 2025-11-24
> > **Response to Reviewer Vwim (part 2)**
> >
> > **W3.** We appreciate the reviewer’s point. Transformer architectures are complex, which makes rigorous theoretical analysis challenging. To make progress, we focused on the fundamental building block—a single self-attention layer. Our theoretical setting and assumptions are consistent with prior work showing fundamental results on transformers [1,2,3,4,5] (e.g., demonstrating that transformers can implement algorithms such as gradient descent, and using bounded input embeddings to obtain generalization bounds for in-context learning [1]). Nevertheless, the insights of our analysis carries over to more complex models, as is confirmed by our experiments with large LVLMs. We further discuss the high-level intuitions below.
> >
> > There is strong support in the literature for why results derived for single-layer or single-head attention extend to deeper and multi-head architectures. The residual stream allows different heads to act as additive, largely independent components operating in approximately orthogonal subspaces, preserving the algorithmic behavior of individual heads without destructive interference [6]. This view aligns with the “ensemble hypothesis,” which proposes that multi-head layers function as collections of semi-independent mechanisms—an interpretation strengthened by empirical findings that many attention heads can be pruned at test time with little performance loss [7].
> >
> > Finally, recent results [8] show that a single self-attention layer can implement one step of gradient descent. Under this perspective, stacking layers corresponds to repeatedly applying this fundamental update rule. Thus, our single-layer analysis is not merely a toy setting but captures the core mechanism that deep transformers iterate throughout the network.
> >
> > Regarding the proxy model, as discussed in lines 172-174, transferability of cross-modal attention matrices between proxy and target LVLMs are shown by prior work (Zhao et al.,
> >
> > 2024), and our empirical results confirm the application of proxy models with various structures (see Figure 3a). Our bounded curvature assumption for fine-tuning is also discussed in prior work (Gekhman et al., 2024; Yang et al., 2024) as we mentioned in lines 208-209. As fine-tuning is short and intends to not change the pretrain parameters substantially (otherwise it harms the performance), its loss is relatively smooth with a bounded curvature.
> >
> > [1] Li, Yingcong, et al. "Transformers as algorithms: Generalization and stability in in-context learning." International conference on machine learning. PMLR, 2023.
> >
> > [2] Song, Bingqing, et al. "Unraveling the gradient descent dynamics of transformers." Advances in Neural Information Processing Systems 37 (2024): 92317-92351.
> >
> > [3] Alman, Josh, and Zhao Song. "Fast attention requires bounded entries." Advances in Neural Information Processing Systems 36 (2023): 63117-63135.
> >
> > [4] Edelman, Benjamin L., et al. "Inductive biases and variable creation in self-attention mechanisms." International Conference on Machine Learning. PMLR, 2022.
> >
> > [5] Weronika Ormaniec, Felix Dangel, and Sidak Pal Singh. What does it mean to be a transformer? International Conference on Learning Representations. ICLR, 2025
> >
> > [6] Elhage, Nelson, et al. "A mathematical framework for transformer circuits." Transformer Circuits Thread 1.1 (2021): 12.
> >
> > [7] Michel, Paul, Omer Levy, and Graham Neubig. "Are sixteen heads really better than one?." Advances in neural information processing systems 32 (2019).
> >
> > [8] Von Oswald, Johannes, et al. "Transformers learn in-context by gradient descent." International Conference on Machine Learning. PMLR, 2023.

---

### Official Review · Reviewer_gorx · 2025-10-28

**Soundness:** 3
**Presentation:** 2
**Contribution:** 2
**Rating:** 4
**Confidence:** 4

**Summary:**

This paper proposes Cross-Modal Alignment SVD (XMAS), a data selection method that identifies and removes redundant training examples in LVLM by clustering samples based on their cross-modal attention trajectories. To achieve this, XMAS fine-tunes a small proxy model and tracks the evolution of  alignment scores for each sample across checkpoints. By sampling a balanced subset of stable samples, XMAS effectively eliminates redundancy without compromising performance.  Experiment results on LLaVA-665K and Vision-Flan datasets demonstrate the effectiveness of the proposed approach.

**Strengths:**

1. This work provides substantial theoretical proofs to elucidate the limitations of existing methods and justify the necessity of the proposed XMAS, which is highly convincing.
2. The paper conducts extensive experiments and comparisons against a wide range of baseline methods, making its findings particularly solid.
3. Experimental results demonstrate that XMAS can achieve nearly identical performance at a reduced training cost, underscoring its effectiveness.

**Weaknesses:**

1. In lines 52-54, the authors state that "the part of the gradient corresponding to multimodal alignment dominates the parts corresponding to individual modalities." Is this claim supported by any prior work or empirical experiments?
2. The writing could be improved, as several informal terms are used (e.g., "big training data").
3. Additional evaluations on diverse Multimodal Large Language Models (MLLMs) could be conducted to further substantiate the generalizability of XMAS.

**Questions:**

Please refer to weakness above.

---

> ### Author Response · Authors · 2025-11-24
> **Response to Reviewer gorx**
>
> We thank the reviewer for recognizing the substantial theoretical results and extensive experiments to verify the effectiveness of our method. Below we would like to address the remaining concerns.
>
> ---
>
> **W1**. As discussed in line 52, image and text embeddings lie in different spaces, and there is a gap between embeddings in the two modalities and the distances between image embeddings, text embeddings, and image-text embeddings have different magnitudes. This is also observed in prior work (Yi et al., 2024; Role et al., 2025). Thus the part of the gradient (captured by attention) corresponding to cross-modal alignment has a different magnitude than the part corresponding to individual modalities. As training mainly aims to align the image and text modalities while preserving the embedding structure within each modality, selecting examples based on cross-modal attention outperforms selection based on full attention. Our ablations in Table 1b further confirm that using cross-modal attention for selection outperforms full-attention.
>
> We have added the above clarification to our revision.
>
> ---
>
> **W2.** We have revised this phrase to “large training datasets” in our revision.
>
> ---
>
> **W3.**
>
> **Diverse proxy models (scale + architecture).** To evaluate robustness to the choice of proxy, we have used proxy models ranging from 0.5B to 2.0B parameters with different vision encoders and language models, as shown in the following (see also Table 2 in Appendix B):
>
> | **Model** | **Language model** | **Vision encoder** |
> | --- | --- | --- |
> | LLaVA-13B | lmsys/vicuna-13b-v1.5 | openai/clip-vit-large-patch14 |
> | LLaVA-7B | lmsys/vicuna-7b-v1.5 | openai/clip-vit-large-patch14 |
> | TinyLLaVA 2.0B | stablelm/stablelm-2-zerphyr-1_6b | openai/clip-vit-large-patch14 |
> | TinyLLaVA 0.5B | Qwen/Qwen2-0.5B | google/siglip-so400m-patch14-384 |
>
> This setup explicitly includes changes in both the LLM backbone (Vicuna vs StableLM vs Qwen) and the vision encoder (CLIP vs SigLIP), not just model scale within a single family.
>
> **Performance under proxy–target mismatch.** We measure how well XMAS preserves performance via the Approximate Ranking Preservation (ARP) metric when using different proxies:
>
> | **Proxy Model** | **Target Model** | **ARP** |
> | --- | --- | --- |
> | LLaVA-13B | TinyLLaVA 2.0B | 95.6 |
> | LLaVA-7B | TinyLLaVA 2.0B | 95.4 |
> | LLaVA-7B | TinyLLaVA 0.5B | 94.3 |
>
> Despite substantial differences in scale, language model, and vision encoder, XMAS achieves impressive ARP across all settings, including the more extreme mismatch case where the proxy is TinyLLaVA-2.0B and the target is LLaVA-13B (c.f. Fig 3a). This indicates that XMAS does not rely on close architectural matching between proxy and target; instead, the alignment trajectories are sufficiently stable across architectures for XMAS to remain effective. As discussed in lines 172-174, transferability of Cross-modal attention matrices between proxy and target LVLMs are also observed by prior work (Zhao et al., 2024).
>
> **Additional target models (New experiments).** To further validate the generalizability of XMAS, we trained two additional target models:
>
> - LLaVA-1.6-Mistral-7B (Vision encoder: openai/clip-vit-large-patch14-336 and LLM decoder: mistralai/Mistral-7B-Instruct-v0.2)
> - Phi-3.5-Vision-Instruct (Vision encoder: openai/clip-vit-large-patch14-336 and LLM decoder: microsoft/Phi-3-mini-128k-instruct)
>
> We use TinyLLaVA 2.0B as the proxy to find 10% of LLaVA 665K with XMAS to train the above target models, and compare XMAS against other methods.
>
> Table below (please refer to Table 6 in our revision for full results of all 4 different target models) shows that XMAS consistently achieves the best average relative performance (ARP) and is the only method that outperforms random selection across all target model architectures. In contrast, COINCIDE relies on hand-picked features, which fail to generalize to these two new models.
>
> | Method | LLaVA-1.6-Mistral-7B | Phi-3.5-Vision-Instruct |
> | --- | --- | --- |
> | Random | 91.9 | 101.8 |
> | Middle Perplexity | 88.8 | 101.5 |
> | Highest Learnability | 78.1 | 102.2 |
> | E2LN | 87.8 | 101.7 |
> | D2 Pruning | 89.5 | 101.7 |
> | SemDeDup | 92.1 | 102.6 |
> | CLIP-Score | 86.7 | 102.3 |
> | Self-Sup | 88.9 | 102.1 |
> | Self-Filter | 88.4 | 102.4 |
> | COINCIDE | 91.6 | 101.9 |
> | XMAS | **93.8** | **103.3** |
>
> We also experimented with Qwen-VL and Intern-VL models, but these models are already strong and do not significantly benefit from fine-tuning on public multimodal datasets such as LLaVA 665K or Vision-Flan. Their base model performance is comparable to that of the fine-tuned versions on these datasets. This reflects a limitation of the available public datasets, rather than a weakness of our method or analysis.

---

> > ### Comment · Reviewer_gorx · 2025-11-24
> >
> > The rebuttal has addressed my concerns, and I have consequently decided to increase my score. However, I would recommend that the authors include relevant references directly within the rebuttal text (e.g., “This phenomenon has also been observed in prior work (Yi et al., 2024; Role et al., 2025)”) to facilitate easier verification by reviewers. Additionally, I encourage the authors to carefully review the writing and grammar in the revised manuscript and make necessary corrections prior to final submission.

---

> ### Author Response · Authors · 2025-11-24
> **Thank you**
>
> Thank you very much for your positive feedback and for increasing your score from 4 to 6. We appreciate your suggestion to include the relevant references directly in the rebuttal text; this addition (in blue in our revision) now appears in **Lines 052–056**. We will also carefully review the writing and grammar throughout the manuscript to ensure clarity and correctness in the final submission.

---

### Official Review · Reviewer_Zjo5 · 2025-10-31

**Soundness:** 3
**Presentation:** 4
**Contribution:** 4
**Rating:** 6
**Confidence:** 3

**Summary:**

This paper addresses the data inefficiency of instruction tuning for Large Vision-Language Models, noting that existing methods often fail to outperform random selection. To solve this, the paper proposes XMAS, a method centered on the idea that data redundancy should be defined by gradient similarity during training. As direct gradient computation is infeasible , XMAS instead trains a small proxy LVLM and tracks each data point's "cross-modal alignment trajectory", defined as the evolution of the top singular values of its cross-modal attention matrix across training checkpoints. By clustering these trajectories and performing a balanced sampling of the most stable examples from these clusters, XMAS effectively removes data redundancy. Experiments demonstrate that this method can discard 50% of the LLaVA-665k dataset and 85% of the Vision-Flan dataset while maintaining full performance , achieving a 1.2x training speedup and significantly outperforming existing baselines.

**Strengths:**

1. XMAS is the only method in the experiment that consistently outperforms random selection across both LLaVA-665k and Vision-Flan datasets, at all tested data budgets (5% to 50%).
2. Achieving full-data performance while dropping 50% of LLaVA-665k and 85% of Vision-Flan  is a remarkable achievement.
3. The method not only improves data efficiency but also delivers a 1.2x end-to-end training speedup, including the selection overhead, making it a genuinely practical tool.

**Weaknesses:**

1. This is the paper's primary weakness. Its theoretical justification relies on an extremely simplified single-layer, single-head, L2-loss model. The authors fail to provide sufficient argument for why this theory should extrapolate to the practical deep, multi-head, cross-entropy-loss LLaVA models. This makes the theoretical section seem more like "post-hoc decoration" than a solid foundation for the method's success.
2. The XMAS method itself introduces new, and seemingly sensitive, hyperparameters: the number of clusters K and the number of checkpoints T. The ablation studies show that the choice of K and T significantly impacts performance. This creates a new problem: users may need to tune K and T before using XMAS, adding to the method's cost and complexity.
3. The method assumes the alignment trajectories of the proxy model (TinyLLaVA-2B) and target model (LLaVA-1.5-7B)  are similar. While Figure 4 provides evidence for this , the experiment is limited to two scales of the same model family (TinyLLaVA). It is unclear if XMAS would remain effective if the proxy and target models had significant architectural differences (e.g., different LLM backbones or Vision Encoders).

**Questions:**

Please respond to the weaknesses I mentioned above.

---

> ### Author Response · Authors · 2025-11-24
> **Response to Reviewer Zjo5 (part 1)**
>
> We thank the reviewer for recognizing the remarkable performance and efficiency of our method. Below we would like to address the remaining concerns.
>
> ---
>
> **W1.** As theoretically analyzing large transformers is infeasible (we’re not aware of any existing theoretical results analyzing deep multi-head cross-entropy-loss LLaVA models), theoretical analyses are often focused on simplified models. The setting of our theoretical analysis and our assumptions is consistent with prior work proving fundamental results on transformers, e.g. for showing transformers can implement algorithms such as gradient descent and for obtaining generalization bounds for in-context-learning [1, 2, 3, 4, 5]. Nevertheless, the insights of our analysis carries over to more complex models, as is confirmed by our experiments with large LVLMs. We further discuss the high-level intuitions below.
>
> There is strong support in the literature for why results derived for single-layer or single-head attention extend to deeper and multi-head architectures. The residual stream allows different heads to act as additive, largely independent components operating in approximately orthogonal subspaces, preserving the algorithmic behavior of individual heads without destructive interference [6]. This view aligns with the “ensemble hypothesis,” which proposes that multi-head layers function as collections of semi-independent mechanisms—an interpretation strengthened by empirical findings that many attention heads can be pruned at test time with little performance loss [7].
>
> Finally, recent results [8] show that a single self-attention layer can implement one step of gradient descent. Under this perspective, stacking layers corresponds to repeatedly applying this fundamental update rule. Thus, our single-layer analysis is not merely a toy setting but captures the core mechanism that deep transformers iterate throughout the network.
>
> [1] Li, Yingcong, et al. "Transformers as algorithms: Generalization and stability in in-context learning." International conference on machine learning. PMLR, 2023.
>
> [2] Song, Bingqing, et al. "Unraveling the gradient descent dynamics of transformers." Advances in Neural Information Processing Systems 37 (2024): 92317-92351.
>
> [3] Alman, Josh, and Zhao Song. "Fast attention requires bounded entries." Advances in Neural Information Processing Systems 36 (2023): 63117-63135.
>
> [4] Edelman, Benjamin L., et al. "Inductive biases and variable creation in self-attention mechanisms." International Conference on Machine Learning. PMLR, 2022.
>
> [5] Weronika Ormaniec, Felix Dangel, and Sidak Pal Singh. What does it mean to be a transformer? International Conference on Learning Representations. ICLR, 2025
>
> [6] Elhage, Nelson, et al. "A mathematical framework for transformer circuits." Transformer Circuits Thread 1.1 (2021): 12.
>
> [7] Michel, Paul, Omer Levy, and Graham Neubig. "Are sixteen heads really better than one?." Advances in neural information processing systems 32 (2019).
>
> [8] Von Oswald, Johannes, et al. "Transformers learn in-context by gradient descent." International Conference on Machine Learning. PMLR, 2023.
>
> ---
>
> **W2.** We thank the reviewer for raising this point. Indeed, XMAS is not very sensitive to the choice of these hyperparameters. For instance, using any number of clusters between 1000 and 2000 consistently works well across datasets, and no additional tuning is necessary. Similarly, for the number of checkpoints, values between 3 and 11 affect the performance by only around 1% and consistently outperform other baselines, with 7 checkpoints achieving the best performance.
>
> To further support this, our experiments on different datasets (LLaVA 665k and VisionFlan in Fig 1 and Fig 2b) **use the same hyperparameters** (1000 clusters, 7 checkpoints) and achieve strong results, without further tuning. We also evaluated XMAS with different combinations of vision encoders and LLM decoders using the same hyperparameters (Fig. 3(a) and Table 2 in Appendix B), which demonstrates the robustness of our method. In comparison, the best baseline, namely COINCIDE, shows substantial sensitivity (2.5%) to the choice of layers (Table 5(c) in Appendix B), highlighting that XMAS is stable with respect to hyperparameters.

---

> > ### Author Response · Authors · 2025-11-24
> > **Response to Reviewer Zjo5 (part 2)**
> >
> > **W3.**
> >
> > **Diverse proxy models (scale + architecture).** To evaluate robustness to the choice of proxy, we have used proxy models ranging from 0.5B to 2.0B parameters with different vision encoders and language models, as shown in the following (see also Table 2 in Appendix B):
> >
> > | **Model** | **Language model** | **Vision encoder** |
> > | --- | --- | --- |
> > | LLaVA-13B | lmsys/vicuna-13b-v1.5 | openai/clip-vit-large-patch14 |
> > | LLaVA-7B | lmsys/vicuna-7b-v1.5 | openai/clip-vit-large-patch14 |
> > | TinyLLaVA 2.0B | stablelm/stablelm-2-zerphyr-1_6b | openai/clip-vit-large-patch14 |
> > | TinyLLaVA 0.5B | Qwen/Qwen2-0.5B | google/siglip-so400m-patch14-384 |
> >
> > This setup explicitly includes changes in both the LLM backbone (Vicuna vs StableLM vs Qwen) and the vision encoder (CLIP vs SigLIP), not just model scale within a single family.
> >
> > **Performance under proxy–target mismatch.** We measure how well XMAS preserves performance via the Approximate Ranking Preservation (ARP) metric when using different proxies:
> >
> > | **Proxy Model** | **Target Model** | **ARP** |
> > | --- | --- | --- |
> > | LLaVA-13B | TinyLLaVA 2.0B | 95.6 |
> > | LLaVA-7B | TinyLLaVA 2.0B | 95.4 |
> > | LLaVA-7B | TinyLLaVA 0.5B | 94.3 |
> >
> > Despite substantial differences in scale, language model, and vision encoder, XMAS achieves impressive ARP across all settings, including the more extreme mismatch case where the proxy is TinyLLaVA-2.0B and the target is LLaVA-13B (c.f. Fig 3a). This indicates that XMAS does not rely on close architectural matching between proxy and target; instead, the alignment trajectories are sufficiently stable across architectures for XMAS to remain effective. As discussed in lines 172-174, transferability of Cross-modal attention matrices between proxy and target LVLMs are also observed by prior work (Zhao et al., 2024).
> >
> > **Additional target models (New experiments)** To further validate the generalizability of XMAS, we trained two additional target models:
> >
> > - LLaVA-1.6-Mistral-7B (Vision encoder: openai/clip-vit-large-patch14-336 and LLM decoder: mistralai/Mistral-7B-Instruct-v0.2)
> > - Phi-3.5-Vision-Instruct (Vision encoder: openai/clip-vit-large-patch14-336 and LLM decoder: microsoft/Phi-3-mini-128k-instruct)
> >
> > We use TinyLLaVA 2.0B as the proxy to find 10% of LLaVA 665K with XMAS to train the above target models, and compare XMAS against other methods.
> >
> > Table below (please refer to Table 6 in our revision for full results of all 4 different target models) shows that XMAS consistently achieves the best average relative performance (ARP) and is the only method that outperforms random selection across all target model architectures. In contrast, COINCIDE (the strongest baseline) relies on hand-picked features, which fail to generalize to these two new models.
> >
> > | Method | LLaVA-1.6-Mistral-7B | Phi-3.5-Vision-Instruct |
> > | --- | --- | --- |
> > | Random | 91.9 | 101.8 |
> > | Middle Perplexity | 88.8 | 101.5 |
> > | Highest Learnability | 78.1 | 102.2 |
> > | E2LN | 87.8 | 101.7 |
> > | D2 Pruning | 89.5 | 101.7 |
> > | SemDeDup | 92.1 | 102.6 |
> > | CLIP-Score | 86.7 | 102.3 |
> > | Self-Sup | 88.9 | 102.1 |
> > | Self-Filter | 88.4 | 102.4 |
> > | COINCIDE | 91.6 | 101.9 |
> > | XMAS | **93.8** | **103.3** |
> >
> > We also experimented with Qwen-VL and Intern-VL models, but these models are already strong and do not significantly benefit from fine-tuning on public multimodal datasets such as LLaVA 665K or Vision-Flan. Their base model performance is comparable to that of the fine-tuned versions on these datasets. This reflects a limitation of the available public datasets, rather than a weakness of our method or analysis.

---

### Official Review · Reviewer_4rJZ · 2025-11-03

**Soundness:** 2
**Presentation:** 2
**Contribution:** 2
**Rating:** 4
**Confidence:** 4

**Summary:**

The paper tackles the problem that LVLMs need huge multimodal instruction-tuning datasets, but a lot of the data is redundant, and existing heuristic data-selection methods (CLIP score, influence, uncertainty, de-dup, activation clustering) don’t reliably beat random selection. The authors argue that redundancy should be defined in terms of gradient similarity—examples that push the model in almost the same direction are redundant—but computing gradient similarity for LVLMs is infeasible. They therefore analyze a (simplified) transformer and show that the distance between two examples’ cross-modal attention matrices upper-bounds their gradient distance, and this relation remains stable across nearby checkpoints (when the Hessian is small, as in instruction tuning). Based on this, they propose XMAS (Cross Modal Alignment SVD): fine-tune a small proxy VLM, track the leading singular values/trajectories of cross-modal attention for each example over a few checkpoints, cluster examples with similar “attention trajectories,” and sample balanced subsets from clusters. XMAS can drop 50% of LLaVA-665k and 85% of Vision-Flan while keeping LLaVA-1.5-7B performance on 10 benchmarks and even speeding training by 1.2×.

**Strengths:**

1.Instead of ad-hoc scores, the paper gives a principled route from “we want to keep gradient-diverse examples” to “we can approximate gradient distance via cross-modal attention distance”. XMAS is designed to be workable: it uses a smaller proxy model and only a few checkpoints, then clusters, so it’s not “compute gradients for every example at every step” unrealistic.

2.Empirical results show that you can drop 50% (LLaVA-665k) or even 85% (Vision-Flan) and match downstream performance — and beat prior data-selection baselines — is a compelling empirical story.

**Weaknesses:**

1.the major contribution of this paper is to use the computation of attention similarity to help select high-quality training examples. As VLm contains a lot of attention layers and attention head, compared with gradient-based data selection strategy, the extraction and computation of attention scores are also memory- and time-consuming. More detailed memory use and time cost for the feature extraction stage should be shown in this work, to highlight the efficiency contribution.

2.In experiment part, the efficiency and effectiveness comparison with baselines like LESS and TIVE, have not been exhibited. In these two papers, they also use LoRA to reduce the memory and computation cost. It might have a lower cost than attention similarity computation in practice.

**Questions:**

Please refer to the questions in weaknesses.

**Details Of Ethics Concerns:**

Please refer to the questions in weaknesses.

---

> ### Author Response · Authors · 2025-11-24
> **Response to Reviewer 4rJZ**
>
> We thank the reviewer for recognizing the principled approach and compelling empirical results of our method. Below we would like to address the remaining concerns.
>
> ---
>
> **W1**. Our method computes attention scores on-the-fly during the model’s forward pass, without requiring any backward computation as in gradient-based strategies. On the other hand, for gradient-based methods, calculating gradients requires backward passes and is very expensive (even with LoRA) for the large training data. Besides, storing the high-dimensional gradient vectors demands considerable memory. As a result, both the time and memory costs are significantly lower than gradient-based ones (e.g. TIVE), demonstrating the efficiency advantage of our approach.
>
> **Memory.** TIVE (which is a gradient-based method) requires training the target model and XMAS trains a much smaller proxy model. During the warm-up training phase, TIVE requires approximately 39 GB of memory on a single GPU, whereas XMAS requires 18 GB for training its smaller proxy, i.e., less than half of TIVE’s memory footprint. For feature extraction, computing alignment scores with XMAS uses about 9 GB of GPU memory, compared to approximately 29 GB for computing gradient features with TIVE, corresponding to less than one-third of TIVE’s memory usage for this step. After extracting alignment scores, XMAS stores only a low-dimensional vector (the alignment trajectory) for each example, requiring just 177 MB of disk space. In contrast, TIVE’s gradients require 21 GB of storage.
>
> **Time.** The table below summarizes the time cost (relative to training the target model on 100% of the data) for selecting a 10% subset using XMAS versus TIVE. Fig 2c also illustrates the timing for different XMAS components.
>
> |  | TIVE | XMAS |
> | --- | --- | --- |
> | Finetuning proxy | 0.08 | 0.18 |
> | Feature extraction (i.e., LoRa gradients vs alignment trajectories) | 0.68 | 0.16 |
> | Clustering + Selection | 0 (422 s) | 0 (1 s) |
> | Finetuning target model | 0.1 | 0.1 |
> | Total | 0.86 | 0.44 |
>
> ---
>
> **W2.**
>
> **LESS**. As we discussed in our related work (titled “Targeted Data Selection”) (Lines 094-095) LESS selects a subset of training data that is most influential for (has the largest gradient similarity to) a particular target downstream data. In contrast, XMAS selects data without being guided by any target task. Therefore, LESS is not a baseline for XMAS.
>
> **TIVE**. This method selects examples that have the largest influence (gradient similarity) to other examples in the same task and selects more from tasks with the smallest average influence (discussed in lines 100-104). However, calculating these gradient similarities (even with LoRA) requires backward passes which is very expensive for the large training data and storing the high-dimensional gradient vectors demands considerable memory (see our response to W1). Moreover, TIVE overlooks the diversity of selected data, which is vital for generalization. In contrast, our approach reduces wall-clock running time (as shown in the table in W1) and guarantees finding a balanced and diverse subset. The table below compares the average relative performance (ARP) of fine-tuning LLaVA-7B on 10% subsets of LLaVA 665k, and confirms the superiority of XMAS.
>
> | Method | ARP* |
> | --- | --- |
> | Random | 93.7 |
> | TIVE | 94.9 |
> | XMAS | 95.6 |
>
> *This ARP excludes the performance on VizWiz because its evaluation server is unavailable during the rebuttal.

---

### Meta-Review · Area_Chair_WRwn · 2026-01-07

**Summary:**

This paper proposes a data selection method, i.e., Cross-Modal Alignment SVD (XMAS), by identifying and removing redundant training examples in LVLM with samples clustered based on their cross-modal attention trajectories. The four reviewers pointed out several critical concerns about different aspects of this paper, including but not limited to:

1. Its theoretical justification relies on an extremely simplified single-layer, single-head, L2-loss model. The authors fail to provide sufficient argument for why this theory should extrapolate to the practical deep, multi-head, cross-entropy-loss LLaVA models. This makes the theoretical section seem more like "post-hoc decoration" than a solid foundation for the method's success.
2. The technical novelty is incremental. The practical pipeline (proxy training + clustering + balanced sampling) remains conceptually close to COINCIDE (activation-based clustering).
3. The XMAS method introduces new, and seemingly sensitive, hyperparameters: the number of clusters K and the number of checkpoints T. How to tune K and T before using XMAS introduces additional cost and complexity.
4. Additional evaluations on diverse Multimodal Large Language Models (MLLMs) could be conducted to further substantiate the generalizability of XMAS.
5. The link between these theoretical results and the practical proxy implementation is assumed rather than demonstrated.
6. As VLm contains a lot of attention layers and attention head, compared with gradient-based data selection strategy, the extraction and computation of attention scores are also memory- and time-consuming. More detailed memory use and time cost for the feature extraction stage should be shown in this work, to highlight the efficiency contribution.
7. Comparison with other baselines is missing, such as the efficiency and effectiveness comparison with baselines like LESS and TIVE.

Some concerns are addressed, such as additional evaluations on diverse MLLMs. However, the concerns about the theoretical justification on simplified settings, incremental technical novelty, and inclusion of more hyperparameters remain challenging.

**Reviewer Concerns:**

The 4th, 5th, and 7th concerns are partially addressed, while the others are still outstanding.

**Reviewer Scores:**

gorx

---

### Decision · Program_Chairs · 2026-01-26

Reject